# Robust Egocentric Referring Video Object Segmentation via Dual-Modal Causal Intervention

**Haijing Liu,    Zhiyuan Song,    Hefeng Wu**,*    **Tao Pu,    Keze Wang,    Liang Lin**\*

Sun Yat-sen University, Guangzhou 510006, China

liuhj66@mail2.sysu.edu.cn, songzhy29@mail2.sysu.edu.cn, wuhefeng@gmail.com,
putao3@mail2.sysu.edu.cn, kezewang@gmail.com, linliang@ieee.org

## Abstract

Egocentric Referring Video Object Segmentation (Ego-RVOS) aims to segment the specific object actively involved in a human action, as described by a language query, within first-person videos. This task is critical for understanding egocentric human behavior. However, achieving such segmentation robustly is challenging due to ambiguities inherent in egocentric videos and biases present in training data. Consequently, existing methods often struggle, learning spurious correlations from skewed object-action pairings in datasets and fundamental visual confounding factors of the egocentric perspective, such as rapid motion and frequent occlusions. To address these limitations, we introduce **C**ausal **E**go-**RE**ferring **S**egmentation (**CERES**), a plug-in causal framework that adapts strong, pre-trained RVOS backbones to the egocentric domain. CERES implements dual-modal causal intervention: applying backdoor adjustment principles to counteract language representation biases learned from dataset statistics, and leveraging front-door adjustment concepts to address visual confounding by intelligently integrating semantic visual features with geometric depth information guided by causal principles, creating representations more robust to egocentric distortions. Extensive experiments demonstrate that CERES achieves state-of-the-art performance on Ego-RVOS benchmarks, highlighting the potential of applying causal reasoning to build more reliable models for broader egocentric video understanding.

## 1 Introduction

Egocentric vision, capturing the world from a first-person perspective, offers invaluable data for understanding human interaction and behavior. Within this domain, Egocentric Referring Video Object Segmentation (Ego-RVOS) [36] presents a key task: segmenting the specific object actively involved in a human action, as identified by a natural language query combining object and action descriptions (Figure 1(a)). Successfully addressing Ego-RVOS paves the way for machines to develop a deeper comprehension of dynamic scenes, integrating visual perception, language understanding, and temporal reasoning. Prior work, such as ActionVOS [36], established settings for this task, notably utilizing action descriptions alongside object names and sometimes adapting pre-trained models with specialized loss functions to focus on active objects.

However, developing robust Ego-RVOS models faces significant hurdles. Current approaches often struggle because they learn spurious correlations rather than genuine cause-and-effect relationships [18, 60, 50, 6, 27]. These spurious correlations stem from two main sources, as illustrated by typical failure cases in Figure 1(b). First, dataset biases often exist where certain object categories frequently co-occur with specific actions, leading models to rely on these statistical shortcuts instead

---

\*Corresponding authors

39th Conference on Neural Information Processing Systems (NeurIPS 2025).

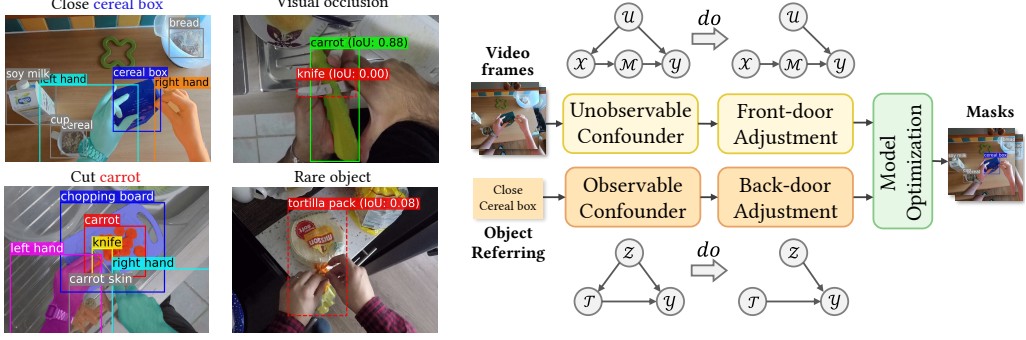

(a) Example of Ego-RVOS (b) Failures due to bias (c) Overview of the CERES causal framework

Figure 1: Motivation and overview of the CERES for addressing biases. (a) Ego-RVOS needs to segment the objects related to action (positive objects, colored) instead of objects unrelated to action (negative objects, gray). (b) Example failure cases of baseline [36] because of visual occlusion and rare objects outside the training set. (c) Our CERES from text and visual modal performs causal intervention to achieve robust Ego-RVOS.

of truly grounding the language query [5, 40, 55]. Second, the inherent nature of egocentric video introduces fundamental visual confounding factors: rapid camera movements, frequent hand-object occlusions, and perspective distortions create complex visual patterns that can mislead models, particularly given the domain shift from typical third-person pre-training data [46, 19, 44, 12, 7, 56]. This reliance on superficial cues renders models brittle and unreliable.

To overcome these challenges and foster robust segmentation, we propose **C**ausal **E**go-**RE**ferring **S**egmentation (CERES), a plug-in causal framework that adapts strong, pre-trained RVOS backbones to the egocentric domain by employing dual-modal causal intervention.

Instead of merely learning correlations, CERES aims to identify and model the underlying causal pathways from the dual-modal inputs (vision $\mathcal{X}$, text $\mathcal{T}$) to the segmentation output ($\mathcal{Y}$), intervening to remove confounding influences. As illustrated in Figure 1(c), we conceptualize the Ego-RVOS process using a causal graph and outline the CERES framework. We identify two primary confounding issues: (1) For the **observable language bias**, stemming from dataset statistics (confounder $\mathcal{Z}$), CERES applies principles inspired by backdoor adjustment [38, 39]. This aims to block the spurious path $\mathcal{T} \leftarrow \mathcal{Z} \rightarrow \mathcal{Y}$ and estimate the direct causal effect of the text query $\mathcal{T}$ on the output $\mathcal{Y}$, $P(\mathcal{Y} \mid \mathrm{do}(\mathcal{T}))$. (2) For the **unobservable visual confounding**, originating from inherent egocentric factors ($\mathcal{U}$), CERES utilizes principles based on **front-door adjustment** [39]. This requires identifying a mediator variable $\mathcal{M}$ that captures the causal effect flowing from vision $\mathcal{X}$ to the output $\mathcal{Y}$. The goal is to estimate the causal effect of the visual input $\mathcal{X}$ on $\mathcal{Y}$, $P(\mathcal{Y} \mid \mathrm{do}(\mathcal{X}))$, by adjusting for the effect mediated through $\mathcal{M}$ while blocking the confounding path $\mathcal{X} \leftarrow \mathcal{U} \rightarrow \mathcal{Y}$.

Implementing the front-door adjustment necessitates a carefully chosen mediator $\mathcal{M}$. Egocentric visual features $\mathcal{X}$ are susceptible to confounding ($\mathcal{U}$) through factors like rapid motion and occlusion. A purely visual mediator risks inheriting this confounding, potentially violating front-door requirements. We hypothesize that incorporating geometric structure can yield a more robust mediator, less sensitive to $\mathcal{U}$. Therefore, we propose a mediator $\mathcal{M}$ integrating semantic visual knowledge ($\mathcal{M}_v$) with geometric depth information ($\mathcal{M}_d$), both derived from $\mathcal{X}$. Leveraging depth cues provides robustness against visual distortions inherent in $\mathcal{U}$, aiming to better isolate the back-door path $\mathcal{X} \rightarrow \mathcal{M} \rightarrow \mathcal{Y}$. CERES employs attention [60] to realize this vision-depth fusion and implement the necessary causal adjustments within an end-to-end framework. The main contributions of this work are:

• We propose CERES, a novel framework applying causal inference principles to tackle key robustness challenges in Ego-RVOS.

• We employ backdoor adjustment concepts to mitigate language biases arising from spurious correlations in object-action dataset statistics.

• We utilize front-door adjustment concepts, implemented via a novel vision-depth mediator, to address fundamental visual confounding inherent in the egocentric perspective.

• Extensive experiments across diverse backbones demonstrate that CERES achieves state-of-the-art performance on VISOR, VOST and VSCOS datasets, significantly improving robustness against both linguistic and visual biases.

## 2 Related Work

• **Referring Video Object Segmentation (RVOS)**   Referring Video Object Segmentation (RVOS) aims to segment the object referred to by a natural language expression throughout a video. Existing RVOS tasks [17, 22, 34, 43, 58] are typically constructed by annotating referring expressions on existing video segmentation benchmarks. These expressions often describe static attributes of a single target object. The recent MeViS dataset [13] introduces motion-based language descriptions for video object segmentation. Various methods have been proposed for RVOS [3, 14, 8, 54, 52, 24]. For instance, SLVP [32] extends RVOS to the VISOR dataset [12], while ActionVOS [36] incorporates action narrations to segment active objects in egocentric videos. Despite these advances, most approaches overlook critical challenges such as occlusion and label bias in referring expressions. In this work, we propose a causal inference-based framework to address these issues, enabling more robust and generalizable RVOS models.

• **Causal Inference**   Causal inference has become an increasingly popular tool for uncovering task causality [39], and has been widely integrated into deep learning systems, especially in vision-language tasks such as image recognition [51, 53, 66, 65, 33], image captioning [61, 25], and visual question answering [23, 60]. A common approach is to apply adjustment techniques to mitigate the influence of confounding variables, with some studies exploring counterfactual reasoning [37, 1, 35]. In this work, we focus on intervention-based methods due to their practicality. However, most existing causal learning frameworks are limited to relatively simple tasks and rarely consider complex embodied settings like RVOS. Moreover, current approaches typically apply either back-door [25, 50, 64, 68] or front-door [28, 60, 61] adjustments independently across modalities, failing to account for both observable and unobservable confounders in a unified manner. Unlike prior works such as GOAT [49], which tackles confounders in vision, language, and action history, we propose the first causal framework tailored for RVOS. Specifically, we introduce a novel front-door adjustment that integrates depth information and adjacent frames features to refine segmentation decisions under occlusion. Meanwhile, we design a back-door blocking strategy to statistically correct biases in referring expressions and action labels. Our approach effectively addresses both visual and linguistic confounding effects, leading to more robust and generalizable Ego-RVOS models.

## 3 Preliminary

**Task Formulation.** The Egocentric Referring Video Object Segmentation (Ego-RVOS) task [36] requires segmenting a specific object instance involved in an action within a first-person video. The input consists of an egocentric video clip, represented as a sequence of frames $\mathbf{X} = \{\mathbf{x}_t\}_{t=1}^{t}$, where $x_t \in \mathbb{R}^{H \times W \times 3}$ is the RGB frame realization at time $t$. Alongside the video, a language query $\mathbf{T}$, realized as $\mathbf{t}_{\text{txt}}$, provides both the object category name and a description of the action (e.g., "knife used to cut carrot").

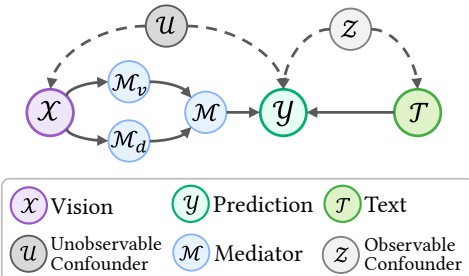

Figure 2: The proposed SCM for Ego-RVOS. (Dashed lines indicate confounding paths; solid lines indicate causal paths.)

The objective is to predict a sequence of binary segmentation masks $\mathbf{Y} = \{\mathbf{y}_t\}_{t=1}^{t}$, where each realization $y_t \in \{0,1\}^{H \times W}$ precisely delineates the pixels belonging to the object instance specified by the query $t$ and actively participating in the described action within frame $x_t$. This task demands robust integration of visual perception ($\mathcal{X}$), language understanding ($\mathcal{T}$), and reasoning about object-action relationships over time.

**Structural Causal Model of Ego-RVOS.** To systematically address the biases inherent in Ego-RVOS, we formulate the task using a Structural Causal Model (SCM) [38], as depicted in Figure 2. This model posits that the visual input $\mathcal{X}$ and text query $\mathcal{T}$ are the direct causes of the segmentation output $\mathcal{Y}$. However, this ideal relationship is often confounded in practice.

We identify two primary confounders. First, an unobserved confounder $\mathcal{U}$ encapsulates intrinsic egocentric visual characteristics (e.g., rapid motion, occlusions) [44]. $\mathcal{U}$ affects both the visual input $\mathcal{X}$ and the output $\mathcal{Y}$, creating a spurious backdoor path $\mathcal{X} \leftarrow \mathcal{U} \rightarrow \mathcal{Y}$. Second, an observable confounder $\mathcal{Z}$ represents dataset-level statistical biases, such as skewed object-action co-occurrences [40]. $\mathcal{Z}$

influences both the text queries $\mathcal{T}$ and the labels $\mathcal{Y}$, forming another backdoor path $\mathcal{T} \leftarrow \mathcal{Z} \rightarrow \mathcal{Y}$. These backdoor paths lead models to learn superficial correlations rather than true causal relationships.

To mitigate the visual confounding from $\mathcal{U}$, we employ the front-door criterion. This involves an intermediate mediator $\mathcal{M}$ that captures the causal effect from $\mathcal{X}$ to $\mathcal{Y}$. Recognizing that purely visual knowledge might still be tainted by $\mathcal{U}$, we propose a more robust mediation strategy. As shown in Figure 2, we conceptualize the visual information $\mathcal{X}$ as giving rise to distinct semantic knowledge $\mathcal{M}_v$ (what objects are present) and geometric depth knowledge $\mathcal{M}_d$ (their spatial layout and structure). We hypothesize that $\mathcal{M}_v$ forms the primary pathway to an intermediate representation $\mathcal{M}$ ($\mathcal{X} \rightarrow \mathcal{M}_v \rightarrow \mathcal{M} \rightarrow \mathcal{Y}$), while $\mathcal{M}_d$ offers a complementary, potentially more robust, pathway influencing $\mathcal{Y}$ ($\mathcal{X} \rightarrow \mathcal{M}_d \rightarrow \mathcal{Y}$). This decomposition aims to leverage the stability of geometric cues ($\mathcal{M}_d$) to buttress the semantic interpretation ($\mathcal{M}_v$), leading to a mediator $\mathcal{M}$ (or a combined influence on $\mathcal{Y}$) that is less susceptible to the distortions introduced by $\mathcal{U}$. The specific mechanisms for realizing these causal adjustments will be detailed in our method section.

# 4 Methodology

The CERES (Causal Egocentric Referring-Segmentation) framework implements distinct causal adjustment strategies to address the language and visual biases inherent in Ego-RVOS, as outlined in our Structural Causal Model (SCM, Figure 2). Specifically, to counteract:

• **Language Bias**: Stemming from the observable confounder $\mathcal{Z}$ in the $\mathcal{T} \leftarrow \mathcal{Z} \rightarrow \mathcal{Y}$ pathway, CERES applies back-door adjustment.

• **Visual Bias**: Originating from the unobserved confounder $\mathcal{U}$ affecting the $\mathcal{X} \leftarrow \mathcal{U} \rightarrow \mathcal{Y}$ pathway, CERES employs front-door adjustment. This is operationalized using a mediator $\mathcal{M}$ which is carefully constructed from semantic visual features ($\mathcal{M}_v$) and geometric depth features ($\mathcal{M}_d$) derived from the visual input $\mathcal{X}$.

The subsequent sections detail the specific formulations for these back-door and front-door adjustments and describe their neural network implementations within CERES.

## 4.1 Language De-biasing via Back-Door Adjustment

Dataset statistics often correlate a textual query $\mathcal{T}$ with its target mask $\mathcal{Y}$ through a visible confounder $\mathcal{Z}$ (e.g., frequent "knife–cut" pairs). Following Pearl's back-door criterion [39], the interventional distribution is

$$P(\mathcal{Y} \mid \mathrm{do}(\mathcal{T} = t)) = \sum_z P(\mathcal{Y} \mid \mathcal{T} = t, \mathcal{Z} = z)\, P(\mathcal{Z} = z) = \mathbb{E}_{\mathcal{Z}}\big[P(\mathcal{Y} \mid t, z)\big]. \tag{1}$$

**Normalized–exponential approximation.** A modern segmenter first maps inputs to pre–activation scores (logits) $s_{\mathcal{Y}}(t, z)$ and then applies the $\mathrm{Softmax}$ function to obtain probabilities. For many practical score distributions, the expectation of Softmax outputs can be closely approximated by applying the Softmax function to the expected scores. This is because for any function of the form $f(z) = \exp(g(z))$, the weighted geometric mean $\prod_z f(z)^{P(z)}$ is equal to $\exp\big(\mathbb{E}_{\mathcal{Z}}[g(z)]\big)$. Given the exponential nature of the Softmax function, this leads to the following approximation (often referred to as the Normalized Weighted Geometric Mean or NWGM approximation [2, 25]):

$$\mathbb{E}_{\mathcal{Z}}\big[\,\mathrm{Softmax}\big(s_{\mathcal{Y}}(t, z)\big)\big] \approx \mathrm{Softmax}\big(\mathbb{E}_{\mathcal{Z}}[s_{\mathcal{Y}}(t, z)]\big). \tag{2}$$

**Additive score assumption.** We further assume the pre-activation scores decompose as $s_{\mathcal{Y}}(t, z) \simeq s_{\mathcal{T}}(t) + s_{\mathcal{Z}}(z)$, a standard design when text and bias features are fused by summation before the final classifier. Substituting into Equation (2) gives the *de-confounded score*:

$$s'_{\mathcal{Y}}(t) = s_{\mathcal{T}}(t) + \mathbb{E}_{\mathcal{Z}}[s_{\mathcal{Z}}(z)]. \tag{3}$$

**Practical estimator.** We instantiate $s_{\mathcal{T}}(t) = \mathbf{w}^\top \mathbf{f}_{\mathcal{T}}(t)$ with a text encoder $\mathbf{f}_{\mathcal{T}}$. A dictionary $\{\mathbf{f}_{\mathcal{Z}}(z_i)\}_{i=1}^K$ of confounder embeddings is built once from the training set (each $z_i$ is a unique object–action pair); the empirical frequency of each $z_i$ serves as $P(z_i)$. The expectation becomes the fixed vector $\bar{\mathbf{f}}_{\mathcal{Z}} = \sum_{i=1}^K P(z_i)\, \mathbf{f}_{\mathcal{Z}}(z_i)$. Finally, the de-biased text representation is

$$\mathbf{f}'_{\mathcal{T}}(t) = \mathbf{f}_{\mathcal{T}}(t) + \bar{\mathbf{f}}_{\mathcal{Z}}, \qquad \text{and } s'_{\mathcal{Y}}(t) = \mathbf{w}^\top \mathbf{f}'_{\mathcal{T}}(t). \tag{4}$$

This implements Equation (3), providing an approximation of $P(\mathcal{Y} \mid \mathrm{do}(\mathcal{T}))$ that is provably back-door adjusted under the stated assumptions.

## 4.2 Visual De-biasing via Front-Door Adjustment

The visual pathway is confounded by an *unobserved* confounder $\mathcal{U}$ (e.g., rapid camera motion, occlusions), rendering a back-door adjustment strategy impossible. Instead, we exploit the two-step mediator process $\mathcal{X} \rightarrow (\mathcal{M}_v, \mathcal{M}_d) \rightarrow \mathcal{M} \rightarrow \mathcal{Y}$ to apply front-door identification, where $\mathcal{M}_v$ represents semantic visual features and $\mathcal{M}_d$ represents geometric features.

**Front-door estimand.** For the causal chain $\mathcal{X} \rightarrow (\mathcal{M}_v, \mathcal{M}_d) \rightarrow \mathcal{M} \rightarrow \mathcal{Y}$, Pearl's front-door theorem yields:

$$P(\mathcal{Y} \mid \mathrm{do}(\mathcal{X} = x)) = \sum_m \sum_{x'} P(\mathcal{Y} \mid \mathcal{M} = m, \mathcal{X} = x') \, P(\mathcal{M} = m \mid \mathcal{X} = x) \, P(\mathcal{X} = x'). \quad (5)$$

To implement this, two key expectations need to be approximated using their feature embeddings (denoted by bold capitals): (i) the effect of general visual context $P(x')$, approximated via $\hat{\mathbf{X}} \approx \mathbb{E}_{\mathcal{X}'}[\mathbf{X}']$, and (ii) the effect of the current visual input $x$ on the mediator $P(m \mid x)$, approximated via $\hat{\mathbf{M}} \approx \mathbb{E}_{\mathcal{M}|\mathcal{X}=x}[\mathbf{M}]$.

**Mediator design.** $\mathcal{M}_v$ encodes high-level semantics from RGB features but can be sensitive to the confounder $\mathcal{U}$. $\mathcal{M}_d$ encodes geometric information (e.g., from pretrained monocular estimation model) and is empirically more robust to $\mathcal{U}$. These are extracted by modality-specific encoders, producing sets of token vectors: $\mathbf{M}_v(x) = \{\mathbf{m}_{v,j}\}_{j=1}^{L_v}$ from the current visual input $x$, and similarly $\mathbf{M}_d(x) = \{\mathbf{m}_{d,k}\}_{k=1}^{L_d}$.

**Self-normalizing token aggregation for mediator components.** To approximate the conditional expectations $\mathbb{E}[\mathbf{M}_v \mid \mathcal{X} = x]$ and $\mathbb{E}[\mathbf{M}_d | \mathcal{X} = x]$ (which contribute to forming $\hat{\mathbf{M}}$), we employ a weighted aggregation of tokens. The weights are derived from a normalized exponential of dot-product similarities between query projections (derived from $x$) and key projections of the tokens:

$$\hat{\mathbf{M}}_v = \sum_{j=1}^{L_v} \frac{\exp(\langle \mathbf{q}_v(x), \mathbf{k}_{v,j}(\mathbf{m}_{v,j}) \rangle)}{\sum_{p=1}^{L_v} \exp(\langle \mathbf{q}_v(x), \mathbf{k}_{v,p}(\mathbf{m}_{v,p}) \rangle)} \, \mathbf{m}_{v,j}, \quad \hat{\mathbf{M}}_d = \sum_{k=1}^{L_d} \frac{\exp(\langle \mathbf{q}_d(x), \mathbf{k}_{d,k}(\mathbf{m}_{d,k}) \rangle)}{\sum_{q=1}^{L_d} \exp(\langle \mathbf{q}_d(x), \mathbf{k}_{d,q}(\mathbf{m}_{d,q}) \rangle)} \, \mathbf{m}_{d,k}, \quad (6)$$

where $\mathbf{q}_*(\cdot)$ and $\mathbf{k}_{*,\cdot}(\cdot)$ represent learned query and key projection functions.

**Cross-modal aggregation for the final mediator $\hat{\mathbf{M}}$.** We assume that the latent mediator $\mathbf{M}$ is a *convex combination* of the visual tokens $\mathbf{M}_v = \{\mathbf{m}_{v,1}, \ldots, \mathbf{m}_{v,L_v}\}$ whose coefficients depend only on the geometric cue $\mathbf{M}_d$. Restricting the predictor to this **attention-linear family (ALF)** yields the set:

$$\mathcal{F}_{\mathrm{ALF}} = \left\{ f_\alpha(\mathbf{M}_v) = \sum_j \alpha_j \mathbf{m}_{v,j} \;\middle|\; \alpha = \mathrm{Softmax}\big(s(\mathbf{M}_d, \mathbf{M}_v)\big) \right\}. \quad (7)$$

With squared error as the loss, the minimum–mean–square–error (MMSE) estimator inside $\mathcal{F}_{\mathrm{ALF}}$ is obtained by choosing the scaled dot-product score $s(\mathbf{M}_d, \mathbf{m}_{v,j}) = \langle W_Q \mathbf{M}_d, W_K \mathbf{m}_{v,j} \rangle / \sqrt{d}$.

Appendix proves this result via a Lagrange-multiplier solution of the simplex-constrained quadratic program [4], following the arguments of Perez et al. [41].

Hence the conditional expectation takes the familiar cross-attention form [48]:

$$\mathbb{E}[\mathbf{M} \mid \mathbf{M}_v(x), \mathbf{M}_d(x)] = \sum_{j=1}^{L_v} \frac{\exp\big(\langle W_Q \mathbf{M}_d(x), W_K \mathbf{m}_{v,j} \rangle\big)}{\sum_{p=1}^{L_v} \exp\big(\langle W_Q \mathbf{M}_d(x), W_K \mathbf{m}_{v,p} \rangle\big)} \, \mathbf{m}_{v,j}. \quad (8)$$

Because $\hat{\mathbf{M}}_d(x) \approx \mathbb{E}[\mathbf{M}_d | \mathcal{X} = x]$ is purposely designed to be insensitive to the egocentric bias $\mathcal{U}$, we obtain a bias-robust estimate of the overall mediator by plugging $\hat{\mathbf{M}}_d(x)$ into Equation (8): $\hat{\mathbf{M}}(x) = \mathrm{Attn}(Q = \hat{\mathbf{M}}_d, \; K = V = \mathbf{M}_v)$. Equation (8) concretely instantiates our claim that "$\mathbf{M}_d$ guides the aggregation of $\mathbf{M}_v$", while ensuring that the learned weights are less exposed to the confounder $\mathcal{U}$. A complete derivation and additional empirical justification are provided in Appendix.

**Memory-bank estimator for $\hat{\mathbf{X}}$.** Most front-door implementations [60, 49] pre-compute a global dictionary of training frames and apply cross-attention against that dictionary to approximate the expectation $\mathbb{E}_{\mathbf{X}'}[\mathbf{X}']$.

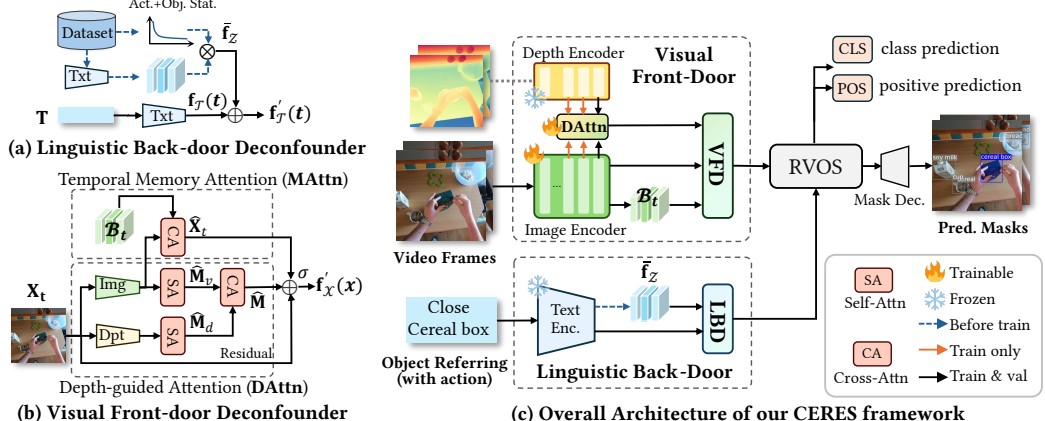

Figure 3: Overview of the CERES framework. The Linguistic Back-door Deconfounder (LBD) de-biases the input text query $\mathbf{T}$ into features $\mathbf{f}'_{\mathcal{T}}(t)$. Concurrently, the Visual Front-door Deconfounder (VFD) processes the video frame $\mathbf{X}_t$; it forms a vision-depth mediator $\hat{\mathbf{M}}(x_t)$ using Depth-guided Attention (DAttn) and estimates temporal visual context $\hat{\mathbf{X}}_t$ via Memory Attention (MAttn), yielding de-biased visual features $\mathbf{f}'_{\mathcal{X}}(x_t)$. These de-biased multimodal features are then used by the RVOS model to predict the segmentation mask $\hat{\mathbf{Y}}_t$.

Such a static lookup is impractical for long, dynamic egocentric streams. We instead assume (A1) short-range stationarity: within a window of $W$ frames, the marginal distribution of the raw frames $x_t$ is approximately unchanged. Under (A1), the last $W$ frames form an i.i.d. Monte-Carlo sample of the stationary distribution, so their empirical average is an unbiased estimate of $\mathbb{E}_{\mathbf{X}'}[\mathbf{X}']$. Concretely, we keep a sliding memory bank $\mathcal{B}_t = \{x_{t-\tau}\}_{\tau=1}^{W}$ and define

$$\hat{\mathbf{X}}_t = \sum_{\tau=1}^{W} \frac{\exp\big(\text{sim}(\mathbf{x}_t, \mathbf{x}_{t-\tau})\big)}{\sum_{\sigma=1}^{W} \exp\big(\text{sim}(\mathbf{x}_t, \mathbf{x}_{t-\sigma})\big)} \mathbf{x}_{t-\tau}, \qquad (9)$$

where $\mathbf{x}_t$ (and $\mathbf{x}_{t-\tau}$) is the frame-level embedding of $x_t$, and $\text{sim}(\cdot, \cdot)$ is a learnable dot-product score as in standard attention [48]. Equation (9) is nothing but a soft weighting of Monte Carlo samples, similar to temperature-scaled importance sampling [63].

**Theoretical guarantee.** By the law of large numbers, the empirical mean over $\mathcal{B}_t$ converges to $\mathbb{E}_{\mathbf{X}'}[\mathbf{X}']$ as $W \to \infty$. Because the softmax weights in Equation (9) satisfy $\sum_{\tau} w_\tau = 1$ and are bounded, the same convergence holds for $\hat{\mathbf{X}}_t$. Combined with the Attention-Linear-Family (ALF) model used for $\hat{\mathbf{M}}$, the overall network is a consistent estimator of the front-door integral.

## 4.3 Overall Architecture

The CERES framework applies our proposed causal adjustment modules, the Linguistic Back-door Deconfounder (LBD) and the Visual Front-door Deconfounder (VFD), to a pre-trained Referring Video Object Segmentation (RVOS) model, typically trained on third-person datasets. An overview of the architecture is depicted in Figure 3.

**Linguistic Back-door Deconfounder (LBD).** To mitigate language bias, the LBD module (Section 4.1) first constructs a confounder dictionary. This dictionary comprises embeddings of unique object-action pairs ($z_i$) and their empirical frequencies $P(z_i)$ derived from the training dataset statistics. The text encoder from the pre-trained RVOS model is used to obtain the initial text query embedding $\mathbf{f}_{\mathcal{T}}(t)$ and the confounder embeddings $\mathbf{f}_{\mathcal{Z}}(z_i)$. During both training and inference, the de-biased text representation $\mathbf{f}'_{\mathcal{T}}(t)$ is computed using Eq. (4), effectively adjusting for spurious correlations learned from dataset statistics.

**Visual Front-door Deconfounder (VFD).** The VFD module (Section 4.2) addresses visual confounding. For each video frame $x_t$, visual features $\mathbf{X}_t^{\text{rgb}}$ are extracted using the image encoder of the pre-trained RVOS model. Concurrently, geometric depth features $\mathbf{X}_t^{\text{depth}}$ are obtained from a pre-trained monocular depth estimation model's encoder.

To construct the mediator component $\hat{\mathbf{M}}(x_t)$, features from the last $n$ layers of both the RGB encoder ($\mathbf{M}_{v,l}(x_t)$ for layer $l$) and depth encoder ($\mathbf{M}_{d,l}(x_t)$ for layer $l$) are utilized. For each of these $n$ layers, **Depth-guided mediator Attention (DAttn)** combines these modalities:

$$\hat{\mathbf{m}}_l(x_t) = \text{DAttn}(Q = \hat{\mathbf{M}}_{d,l}(x_t), K = V = \mathbf{M}_{v,l}(x_t)), \qquad (10)$$

Table 1: Comparison (%) with state-of-the-art methods on VISOR. Best results are highlighted in **bold**, second-best in underline within the same backbone. ↑ indicates higher is better, ↓ indicates lower is better.

| Method | Backbone | mIoU$^{\oplus}$↑ | cIoU$^{\oplus}$↑ | mIoU$^{\ominus}$↓ | cIoU$^{\ominus}$↓ | gIoU↑ | Acc↑ | F1↑ |
|---|---|---|---|---|---|---|---|---|
| ReferFormer | R101 | 59.9 | 66.4 | 30.5 | 52.1 | 55.3 | 58.6 | 64.2 |
| ReferFormer+ | R101 | 58.2 | 64.8 | **14.3** | **18.9** | 63.1 | 67.6 | 68.9 |
| ActionVOS | R101 | 59.9 | 67.2 | 16.3 | 28.5 | 69.9 | 73.4 | 73.7 |
| Ours | R101 | **64.0** | **72.8** | 15.3 | 25.6 | **72.4** | **76.3** | **77.1** |
| ReferFormer+ | VSwinB | 61.1 | 68.9 | 19.2 | 36.8 | 68.4 | 73.2 | 74.0 |
| ActionVOS | VSwinB | 62.9 | 70.9 | 20.0 | 38.8 | 69.5 | 70.7 | 74.3 |
| Ours | VSwinB | **65.4** | **72.5** | **19.1** | **35.1** | **72.1** | **74.7** | **75.9** |
| HOS | SwinL | 55.1 | 59.2 | **13.5** | **17.3** | 66.5 | 70.3 | 69.7 |
| ActionVOS | SwinL | 66.3 | 71.9 | 22.8 | 42.5 | 68.7 | 73.4 | 75.5 |
| Ours | SwinL | **67.0** | **73.6** | 16.9 | 28.6 | **71.8** | **75.2** | **76.2** |

Table 2: Comparison (%) with state-of-the-art methods on the novel subset of VISOR. Best results are in **bold**, second-best in underline. ↑ indicates higher is better, ↓ indicates lower is better.

| Method | mIoU$^{\oplus}$↑ | cIoU$^{\oplus}$↑ | mIoU$^{\ominus}$↓ | cIoU$^{\ominus}$↓ | gIoU↑ | Acc↑ | F1↑ |
|---|---|---|---|---|---|---|---|
| ReferFormer+ | 47.2 | 54.1 | 13.5 | 21.8 | 50.1 | 51.9 | 57.7 |
| HOS | 45.8 | 49.7 | **7.8** | **11.1** | 61.9 | 64.5 | 66.0 |
| ActionVOS | 55.3 | 62.8 | 14.5 | 25.4 | 65.8 | 69.4 | 71.9 |
| Ours | **60.0** | **69.9** | 14.4 | 20.4 | **67.9** | **72.2** | **75.8** |

where $\hat{\mathbf{M}}_{d,l}(x_t)$ is the aggregated depth feature representation for layer $l$ (analogous to Eq. (6) applied layer-wise). This yields layer-specific mediator representations $\{\hat{\mathbf{m}}_l(x_t)\}_{l=1}^n$. For the final mask prediction, we typically use the mediator from the last processed layer, $\hat{\mathbf{M}}(x_t) = \hat{\mathbf{m}}_n(x_t)$.

To estimate the general visual context $\hat{\mathbf{X}}_t$, a memory bank stores recent frame features (from $\mathbf{X}_{\text{rgb}}$), augmented with temporal positional encodings. **Temporal Memory Attention (MAttn)** then computes $\hat{\mathbf{X}}_t$ as per Eq. (9). The de-biased visual feature $\mathbf{f}'_{\mathcal{X}}(x_t)$ is then formed by integrating the mediator and context information. Specifically, the final mediator $\hat{\mathbf{M}}(x_t)$ and the context $\hat{\mathbf{X}}_t$ are concatenated and processed through an MLP followed by a gated residual connection:

$$\mathbf{f}'_{\mathcal{X}}(x_t) = \sigma_{\mathbf{M},\mathbf{X}} \cdot \text{MLP}([\hat{\mathbf{M}}(x_t); \hat{\mathbf{X}}_t]) + (1 - \sigma_{\mathbf{M},\mathbf{X}}) \cdot \mathbf{X}_t. \tag{11}$$

This $\mathbf{f}'_{\mathcal{X}}(x_t)$ represents the visual input adjusted for egocentric confounders, $\sigma_{\mathbf{M},\mathbf{X}}$ is the gate of residual path for easier to train [45, 20]. For auxiliary losses during training, similar de-biased features $\mathbf{f}'_{\mathcal{X},l}(x_t)$ are computed using the intermediate layer-specific mediators $\hat{\mathbf{m}}_l(x_t)$.

**Output Generation and Training.** The de-biased text features $\mathbf{f}'_{\mathcal{T}}(t)$ and visual features $\mathbf{f}'_{\mathcal{X}}(x_t)$ (or $\mathbf{f}'_{\mathcal{X},l}(x_t)$ for intermediate layers) are then utilized by the subsequent components of the RVOS model. These typically include a classification head to predict the object category, a "positive" head to identify if the object is actively involved in the action, and a mask decoder to generate the final segmentation mask $\hat{y}_t$. The model is trained using a standard segmentation loss. Following prior work, auxiliary segmentation losses are applied to the outputs derived from the de-biased visual features of the $n$ intermediate layers during training. During inference, only the de-biased visual feature from the final considered layer is used for prediction.

## 5 Experiment

### 5.1 Experiment Settings

• **Datasets.** Following previous work [36], we evaluate our method on three public egocentric video datasets: VISOR [12], VOST [46], and VSCOS [62]. VISOR, derived from EPIC-KITCHENS [10, 11], provides annotations for hands and active object interactions; we utilize its training and validation splits. After pre-processing, this yields 13,205 videos (76,873 objects) for training and 467 videos (1,841 objects) for validation, where validation objects are manually annotated as positive or negative. VOST and VSCOS are used for validation only. VOST assesses performance on objects undergoing transformations. VSCOS focuses on state-changing objects; its validation data is filtered to prevent overlap with the VISOR training set.

- **Metrics.** We follow established evaluation protocols [26, 36]. Key metrics include mean Intersection over Union (mIoU) and cumulative IoU (cIoU), reported separately for positive (mIoU$^\oplus$, cIoU$^\oplus$) and negative (mIoU$^\ominus$, mIoU$^\ominus$) objects to assess segmentation of interacted and non-interacted instances, respectively. We also report generalized IoU (gIoU) [26] for a combined assessment of segmentation and target classification. Additionally, Precision (P), Recall (R), and Accuracy (Acc) are used for the binary classification of object activity. For these classification-related evaluations (gIoU, P, R, Acc), a prediction is considered a True Positive (TP) if its IoU with the ground truth exceeds 0.5. More metrics details are in the Appendix D.

- **Implementation Details.** CERES builds upon the ReferFormer [57] architecture, initializing with its pre-trained weights. We evaluate ResNet101 [20], Swin-Transformer-L [30], and Video Swin-Transformer-B [31] as image encoder backbones, paired with a RoBERTa [29] text encoder. Input frames are resized to $448 \times 448$. For the Depth-guided Attention (DAttn) of Visual Front-door Deconfounder (VFD), depth features are extracted using the encoder of a frozen pre-trained Depth Anything V2 model [59]. The Linguistic Back-door Deconfounder (LBD) defines confounders $z_i$ based on unique "verb-noun" pairs identified in the training set queries. The VFD's Temporal Memory Attention (MAttn) employs a window of $W = 5$ recent frames. The model is trained with a batch size of 4. Following the previous implement, auxiliary losses during training utilize visual features from the last three layers of the image encoder. During inference, predictions are made **online**, without access to future frames. Further optimization and hyperparameter details are provided in the Appendix.

## 5.2 Comparison with State-of-the-art Methods

We compare our CERES against several leading methods. ReferFormer [57] serves as a foundational baseline, representing a model pre-trained on third-person data and subsequently fine-tuned for Ego-RVOS. ReferFormer+ extends this by incorporating an auxiliary prediction head for "positiveness" (identifying if the object is actively involved in the action) and by including action descriptions in the referring query. EgoHOS [67] is a hand-object segmentation model; for our comparison, we train it on VISOR, treating hand-associated objects as positive targets. ActionVOS [36] is a strong recent baseline that also builds on pre-trained models and employs specialized losses for active objects.

Table 3: Comparison (%) of positive objects IoU on VSCOS and VOST.

| Method | VSCOS | | VOST | |
|---|---|---|---|---|
| | mIOU↑ | cIOU↑ | mIOU↑ | cIOU↑ |
| ReferFormer+ | 53.0 | 54.2 | 30.6 | 16.1 |
| HOS | 42.1 | 31.2 | 21.9 | 15.2 |
| ActionVOS | 52.5 | 57.7 | 30.2 | 17.6 |
| Ours | **55.3** | **62.5** | **32.0** | **21.7** |

On the VISOR benchmark (Table 1), CERES consistently outperforms prior methods across all backbones. With ResNet101, CERES achieves 64.0% mIoU$^\oplus$ (+4.1% over ActionVOS) and 72.4% gIoU (+2.5%), alongside improved accuracy and F1 scores for positive object classification. CERES also generally shows lower mIoU$^\ominus$ and cIoU$^\ominus$, indicating better discrimination against non-target objects. While HOS has low mIoU$^\ominus$ due to its focus on hand-proximate objects, its mIoU$^\oplus$ is consequently limited.

CERES's advantages are particularly evident on a VISOR subset with novel objects or actions not seen during training (Table 2). This subset serves as an open-vocabulary evaluation within VISOR, and CERES significantly surpasses previous state-of-the-art results, demonstrating strong generalization capabilities likely due to its mitigation of dataset-induced language biases.

Evaluations on VSCOS and VOST (Table 3), which feature significant object transformations, further underscore CERES's robustness. We treat these as zero-shot open-vocabulary generalization: models are trained only on VISOR and evaluated on VSCOS/VOST without any fine-tuning. For instance, on VSCOS, CERES achieves 55.3% mIoU$^\oplus$ and 62.5% cIoU$^\oplus$, exceeding ActionVOS (52.5% and 57.7%). This superior performance in challenging scenarios highlights the effectiveness of CERES's causal intervention strategies in addressing both linguistic and visual confounding factors in egocentric videos.

## 5.3 Ablation Study

We conduct ablation studies on the VISOR dataset using the ResNet101 backbone to analyze the contribution of each key component in CERES.

**Ablation of Proposed Modules.** In Table 4, the baseline model (first row) achieves 59.9% mIoU$^\oplus$ and 69.9% gIoU. Introducing only the LBD (second row) improves mIoU$^\oplus$ to 61.2% (+1.3%) and gIoU to 71.4% (+1.5%), demonstrating its effectiveness in mitigating language bias. A nonlinear MLP-based depth fusion (third row) for the mediator offers a 62.1% mIoU$^\oplus$ but increases mIoU$^\ominus$, indicating limited discriminative benefit. In contrast, our DAttn depth integration (fourth row) significantly boosts mIoU$^\oplus$ to 63.3% (+3.4% over baseline) and reduces mIoU$^\ominus$ to 15.8%, yielding substantial gains in gIoU (71.8%) and Acc (75.3%). This confirms

Table 4: Ablation study (%) of proposed modules on VISOR (ResNet101). (◇ indicates MLP-based depth fusion)

| DAttn | MAttn | LBD | mIoU$^{\oplus}$↑ | mIoU$^{\ominus}$↓ | gIoU↑ | Acc↑ |
|---|---|---|---|---|---|---|
| | | ✓ | 59.9 | 16.3 | 69.9 | 73.4 |
| | | ✓ | 61.2 | 16.0 | 71.4 | 74.8 |
| ◇ | | | 62.1 | 17.5 | 70.5 | 73.6 |
| ✓ | | | 63.3 | 15.8 | 71.8 | 75.3 |
| ✓ | ✓ | | 63.1 | **14.9** | 72.1 | 76.1 |
| ✓ | ✓ | ✓ | **64.0** | 15.3 | **72.4** | **76.3** |

Table 5: Performance comparison on a "hard" subset of VISOR. (RF means ReferFormer [57], ActV means ActionVOS [36])

| Method | mIoU$^{\oplus}$↑ | mIoU$^{\ominus}$↓ | gIoU↑ | Acc↑ |
|---|---|---|---|---|
| RF | 53.4 | 14.9 | 56.3 | 58.5 |
| RF+ | 54.2 | 14.7 | 56.7 | 58.7 |
| ActV | 58.4 | 15.1 | 69.9 | 73.1 |
| Ours | **62.3** | **14.3** | **72.2** | **75.6** |

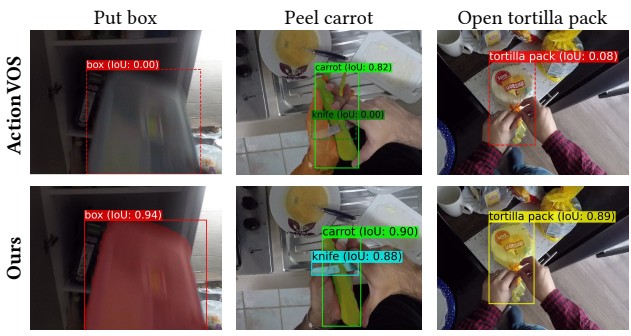

Figure 4: Qualitative analysis of ActionVOS and our CERES.

Figure 5: Effect of $W$ in MAttn.

the superiority of our causally-inspired depth mediator design within the VFD. Adding MAttn (fifth row) to complete the VFE (DAttn + MAttn) further refines performance. This configuration achieves the lowest mIoU$^{\ominus}$ (14.9%) and a strong 72.1% gIoU, highlighting TCA's role in improving discrimination by modeling broader visual context. The full CERES model (last row), integrating both the complete VFE and LBD, attains the best overall performance, with 64.0% mIoU$^{\oplus}$, 72.4% gIoU, and 76.3% Acc. While its mIoU$^{\ominus}$ (15.3%) is slightly higher than VFE-only (14.9%), the LBD's inclusion enhances the recall of positive instances. This trade-off results in superior overall identification and segmentation of target objects. These ablations validate the individual and synergistic contributions of our causal adjustment modules.

**Temporal Context Window Size for MAttn.** We analyze the impact of the temporal window size $W$ for the MAttn module in Figure 5. Setting $W = 0$ (i.e., no MAttn module) results in lower performance compared to using MAttn. As $W$ increases, mIoU$^{\oplus}$ and Acc generally improve. We found $W = 5$ provides a robust balance and consistently strong results.

**Performance on Rare Concepts.** To assess robustness against data scarcity biases, we evaluate models on a "hard" subset of VISOR, comprising 159 clips with actions or objects appearing <50 times in training. Table 5 shows that, compared to **ActionVOS**, **CERES** improves mIoU$^{\oplus}$ by +3.9% (62.3% vs 58.4%) and gIoU by +2.3% (72.2% vs 69.9%). These gains are mainly attributed to **LBD** blocking the spurious path $\mathcal{T} \leftarrow \mathcal{Z} \rightarrow \mathcal{Y}$ (back-door), while **VFD** mitigates egocentric visual confounders $\mathcal{U}$ via the vision–depth mediator (front-door). This highlights the effectiveness of CERES, particularly the LBD component, in generalizing to less frequent concepts by mitigating reliance on spurious statistical correlations.

**Qualitative Analysis.** Figure 4 qualitatively compares CERES with ActionVOS [36], showcasing CERES's superior robustness. Our method yields more accurate segmentation in challenging egocentric scenarios, including those with visual distortions like motion blur (e.g., "box", column 1) and occlusions (e.g., "knife", column 2). This highlights the VFD module's effectiveness in mitigating visual biases. Furthermore, CERES demonstrates improved handling of textual queries involving uncommon objects or actions (e.g., "tortilla pack", column 3), where ActionVOS may falter due to dataset biases. This underscores the LBD module's contribution to better language grounding. Overall, these visual examples corroborate our quantitative results, illustrating how CERES's causal interventions lead to more robust Ego-RVOS.

## 6 Conclusion

This paper introduced Causal Ego-REferring Segmentation (CERES), a novel framework that applies causal inference principles to address critical robustness challenges in Egocentric Referring Video Object Segmentation. We identified two primary sources of error: language biases stemming from dataset statistics and visual confounding inherent in the egocentric perspective. CERES tackles these by employing backdoor adjustment to

mitigate spurious correlations between textual queries and segmentation outputs, and by utilizing front-door adjustment with a novel vision-depth mediator to counteract the effects of unobserved visual confounders. This dual-pronged causal intervention allows CERES to learn more robust representations, less susceptible to dataset-specific biases and egocentric visual distortions. Extensive experiments on standard Ego-RVOS benchmarks demonstrate that CERES achieves state-of-the-art performance, significantly improving segmentation accuracy and reliability, particularly in challenging scenarios with novel concepts or significant visual ambiguity. Our work underscores the potential of causal reasoning to build more generalizable and trustworthy models for complex egocentric video understanding tasks.

## Acknowledgments

This work was supported by National Key R&D Program of China under Grant No. 2021ZD0111601, National Natural Science Foundation of China (NSFC) under Grant No. 62272494, Guangdong Basic and Applied Basic Research Foundation under Grant No. 2023A1515012845 and 2023A1515011374, and Guangdong Province Key Laboratory of Information Security Technology.

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

# Appendix

## A    Limitations

While CERES demonstrates notable advancements in robust Ego-RVOS, we acknowledge certain aspects for future consideration. The LBD module's current definition of confounders ($\mathcal{Z}$) as object-action pairs, while effective for the targeted dataset biases, represents one specific strategy; future work could explore more nuanced or automatically discovered confounder definitions. The additive score assumption used in the back-door adjustment, though common, is an approximation of the true deconfounded score.

For the VFD, its performance is influenced by the capabilities of the underlying pre-trained depth estimator. While we empirically show that depth aids in mitigating visual confounding, formally verifying full front-door conditions for the chosen mediator structure is nontrivial. The Attention-Linear Family (ALF) assumption explicitly trades expressivity for identifiability: restricting fusion weights to be functions of $M_d$ helps prevent leakage of the unobserved confounder $U$ and yields a minimal, identifiable realization of the mediator, but narrows the function class. Empirically, our ALF-based DAttn outperforms a nonlinear MLP fusion (Table 4), indicating superior robustness under egocentric confounding. Exploring richer mediator parameterizations that preserve front-door validity is left for future work.

The MAttn module approximates temporal context using a sliding window, a technique effective for many dynamic egocentric scenes, though its generalization to scenarios with extremely long-range temporal dependencies could be further investigated. The integration of depth features and additional attention mechanisms introduces computational costs relative to simpler baselines, a common trade-off for enhanced robustness, and further optimization could be explored.

Finally, the current evaluation of CERES is focused on the Ego-RVOS task. Extending and rigorously evaluating its applicability and potential adaptations for a broader spectrum of egocentric video understanding challenges, such as egocentric action recognition or long-term activity understanding, presents a valuable avenue for future research.

## B    Broader Impact

The advancements in robust Ego-RVOS achieved by CERES hold potential for significant positive impacts. More reliable egocentric video understanding can directly benefit assistive technologies, enhancing contextual awareness for individuals with visual impairments, and can enable more intuitive human-robot interaction by allowing machines to better grasp human-object interactions from a first-person view. In fields like augmented reality, precise segmentation of actively manipulated objects can lead to more seamless and responsive user experiences. Crucially, the causal principles and architectural components developed in CERES, particularly the strategies for mitigating dataset and visual biases, may offer a foundational approach for enhancing robustness and generalizability in other egocentric video analysis tasks. This is particularly relevant for embodied AI, where a nuanced understanding of human actions and object interactions from an egocentric perspective is critical for agents to learn from human demonstrations, predict intentions, and operate safely and intelligently within complex human environments.

However, as with any technology capable of detailed scene analysis, responsible development is paramount. The potential for misuse in surveillance or intrusive monitoring, especially with personal egocentric data, necessitates strong ethical guidelines and privacy-preserving measures. While CERES aims to reduce specific biases, the underlying pre-trained models might still harbor unaddressed biases, emphasizing the need for continuous auditing and fairness considerations in AI system development. The deployment of such technologies should thus proceed with a commitment to ethical practices and ongoing research into comprehensive bias mitigation.

## C    More Implementation Details

All experiments were conducted using PyTorch 2.1.2 and CUDA 11.8 on a system with four NVIDIA V100 GPUs. Models were trained for 6 epochs with a total batch size of 4, where each batch item was a single video clip. We initialized the learning rate to $1 \times 10^{-3}$ for our CERES modules and $1 \times 10^{-4}$ for pre-trained components, decaying it by 0.1 at epochs 3 and 5, using the AdamW optimizer. The primary segmentation loss combined bounding box, Dice and Focal losses. Input frames were resized to $448 \times 448$ for both training and inference.

We use ReferFormer pretrained weight on Youtube-VOS dataset. For the VFD, depth features were extracted using a frozen pre-trained Depth Anything V2 encoder. The LBD module defined confounders $\mathcal{Z}$ from unique "verb-noun" pairs in training queries. The VFD's DAttn utilized features from the last three layers of the image and depth encoders, while MAttn employed a temporal window of $W = 5$ recent frames. During inference, an online strategy was adopted, processing frames sequentially without access to future information.

## D  Metric Details

This section provides further details on the evaluation metrics used in this work, complementing the descriptions in Section 5.1 of the main paper.

• **Accuracy and F1-score.** These metrics evaluate the binary classification of whether an object is actively involved in the queried action. A prediction is considered a True Positive (TP) if its Intersection over Union (IoU) with the ground-truth active object mask exceeds 0.5. This stricter threshold, compared to some prior works like ActionVOS [36] which might use IoU > 0, ensures a more accurate reflection of identification and localization. True Negatives (TN) are correctly identified non-active objects or correct "no target" predictions. False Positives (FP) are non-active objects misclassified as active, or incorrect "target present" predictions. False Negatives (FN) are active objects missed or misclassified as non-active.

Based on these, Accuracy (Acc) and F1-score are calculated as:

- Accuracy (Acc):

$$\text{Acc} = \frac{\text{TP} + \text{TN}}{\text{TP} + \text{TN} + \text{FP} + \text{FN}}$$

- F1-score:

$$\text{F1} = \frac{2 \times \text{TP}}{2 \times \text{TP} + \text{FP} + \text{FN}}$$

The F1-score is the harmonic mean of Precision and Recall, providing a balanced measure for active object classification.

• **Mean Intersection over Union (mIoU) and Cumulative IoU (cIoU).** These are standard segmentation quality metrics. **mIoU** calculates the average IoU across all individual object instances in the dataset. For each instance, IoU is the ratio of the area of overlap between the predicted mask and the ground-truth mask to the area of their union. **cIoU** computes a single IoU value over the entire dataset (or a subset) by summing all intersection areas and dividing by the sum of all union areas. As stated in the main paper, these metrics are reported separately for positive objects (target objects actively involved in the action: mIoU$^{\oplus}$, cIoU$^{\oplus}$) and negative objects (other objects present in the scene but not involved in the queried action: mIoU$^{\ominus}$, cIoU$^{\ominus}$). Lower scores for negative objects indicate better discrimination against non-targets.

• **Generalized IoU (gIoU).** Proposed by Liu et al. [26], gIoU offers a combined assessment of both segmentation quality and the model's ability to correctly classify target presence. It is calculated as the mean of per-sample scores, making it less sensitive to object size variations compared to cIoU. The per-sample score for gIoU is determined as follows:

- If a ground-truth target object exists for the query in a given sample: The prediction for this object is first evaluated against the TP criterion (IoU > 0.5 with the ground truth). If it qualifies as a TP, its actual IoU value contributes to the gIoU average for that sample. If the prediction's IoU is $\leq 0.5$, or if no object is predicted by the model, the contribution for that sample is 0.

- If the ground truth indicates no target object for the query in a given sample (a "no-target" sample): If the model correctly predicts that no target object is present, the sample's contribution to gIoU is 1. If the model incorrectly predicts an object, the contribution is 0.

This definition ensures that gIoU comprehensively evaluates the model's performance in segmenting correctly identified targets as well as its ability to correctly handle scenarios where the queried object is absent.

## E  Theoretical Details

### E.1  Proof of the Cross–Modal MMSE Estimator

This appendix provides the technical details that underpin Eq. (8) in the main text, showing that the *scaled–dot–product cross-attention*

$$\text{Attn}(Q = \mathbf{M}_d,\ K = V = \mathbf{M}_v)\ =\ \sum_{j=1}^{L_v} \frac{\exp\big(\langle W_Q \mathbf{M}_d,\ W_K \mathbf{m}_{v,j}\rangle / \sqrt{d}\big)}{\sum_p \exp\big(\langle W_Q \mathbf{M}_d,\ W_K \mathbf{m}_{v,p}\rangle / \sqrt{d}\big)}\ \mathbf{m}_{v,j}, \qquad \text{(E.1)}$$

is the *minimum–mean–square–error* (MMSE) estimator of the latent mediator $\mathbf{M}$ *within* the attention–linear family $\mathcal{F}_{\text{ALF}}$ defined in Sec. 4.2. The derivation follows the general recipe of simplex–constrained quadratic optimization [4] and re-uses the causal decomposition arguments of Perez et al. [41]. Throughout the appendix all expectations are conditional on the observed pair $(\mathbf{M}_d, \mathbf{M}_v)$ unless stated otherwise.

### E.1.1 Problem Set-Up

Let $\mathbf{M} \in \mathbb{R}^d$ be the *latent* mask mediator that triggers the downstream decoder. For a fixed input we observe

$$\mathbf{M}_d \in \mathbb{R}^d, \qquad \mathbf{M}_v = \{\mathbf{m}_{v,1}, \dots, \mathbf{m}_{v,L_v}\} \subset \mathbb{R}^d.$$

Inside the family $\mathcal{F}_{\mathrm{ALF}}$ every candidate estimator is a convex combination $f_{\boldsymbol{\alpha}}(\mathbf{M}_v) = \sum_j \alpha_j \mathbf{m}_{v,j}$ with non-negative coefficients $\boldsymbol{\alpha} \in \Delta^{L_v}$, where $\Delta^{L_v} := \{\alpha_j \geq 0, \ \sum_j \alpha_j = 1\}$ is the probability simplex. The **restricted MMSE problem** is therefore

$$\boldsymbol{\alpha}^\star = \operatorname{argmin}_{\boldsymbol{\alpha} \in \Delta^{L_v}} \mathcal{L}(\boldsymbol{\alpha}), \quad \mathcal{L}(\boldsymbol{\alpha}) := \mathbb{E}\big[\|\mathbf{M} - \sum_j \alpha_j \mathbf{m}_{v,j}\|_2^2\big]. \tag{E.2}$$

### E.1.2 Quadratic Form of the Risk

Let $\boldsymbol{\mu} := \mathbb{E}[\mathbf{M}]$ be the (unknown) conditional mean of the latent mask. Expanding the square in Eq. (E.2) gives

$$\mathcal{L}(\boldsymbol{\alpha}) = \underbrace{\|\boldsymbol{\mu}\|^2}_{\text{const}} - 2\,\boldsymbol{b}^\top \boldsymbol{\alpha} + \boldsymbol{\alpha}^\top G\,\boldsymbol{\alpha}, \tag{E.3}$$

where $b_j = \langle \boldsymbol{\mu}, \mathbf{m}_{v,j} \rangle$ and $G_{ij} = \langle \mathbf{m}_{v,i}, \mathbf{m}_{v,j} \rangle$. Because the first term does not depend on $\boldsymbol{\alpha}$, Eq. (E.2) reduces to the *simplex-constrained quadratic program*

$$\min_{\boldsymbol{\alpha} \in \Delta^{L_v}} \big\{ -2\,\boldsymbol{b}^\top \boldsymbol{\alpha} + \boldsymbol{\alpha}^\top G\boldsymbol{\alpha} \big\}. \tag{E.4}$$

### E.1.3 Lagrange–Multiplier Solution

We solve Eq. (E.2) by introducing a Lagrangian

$$\mathcal{J}(\boldsymbol{\alpha}, \lambda, \boldsymbol{\eta}) = -2\,\boldsymbol{b}^\top \boldsymbol{\alpha} + \boldsymbol{\alpha}^\top G\boldsymbol{\alpha} + \lambda\Big(\sum_j \alpha_j - 1\Big) - \boldsymbol{\eta}^\top \boldsymbol{\alpha}, \tag{E.5}$$

where $\lambda \in \mathbb{R}$ and $\boldsymbol{\eta} \in \mathbb{R}_{\geq 0}^{L_v}$ are dual variables that enforce the simplex constraints. Differentiating Eq. (E.2) and using the KKT conditions [4, Ch. 5] yields

$$2\big(G\boldsymbol{\alpha}\big)_j - 2b_j + \lambda - \eta_j = 0, \quad \forall j, \tag{E.6a}$$

$$\eta_j\,\alpha_j = 0, \quad \alpha_j \geq 0, \ \eta_j \geq 0, \tag{E.6b}$$

$$\sum_j \alpha_j = 1. \tag{E.6c}$$

When $G \succ 0$ (true after layer normalization [48]), the solution lies in the *open* simplex—namely $\alpha_j > 0$, implying $\eta_j = 0$. Subtracting the $i$-th and $j$-th rows of Eq. (E.2) cancels $\lambda$ and gives $b_j - b_i = (G\boldsymbol{\alpha})_j - (G\boldsymbol{\alpha})_i$. Re-ordering yields

$$\alpha_j = \alpha_i \exp\Big(\tfrac{b_j - b_i - (G\boldsymbol{\alpha})_j + (G\boldsymbol{\alpha})_i}{\tau}\Big), \tag{E.7}$$

where we have inserted an infinitesimal *temperature* $\tau > 0$ for differentiability (the same trick as entropic regularization [9]). Imposing $\sum_j \alpha_j = 1$ converts Eq. (E.2) into the softmax-style fixed-point

$$\alpha_j = \frac{\exp\big(\tfrac{2b_j - 2(G\boldsymbol{\alpha})_j}{\tau}\big)}{\sum_p \exp\big(\tfrac{2b_p - 2(G\boldsymbol{\alpha})_p}{\tau}\big)}. \tag{E.8}$$

### E.1.4 Isotropic-Token Approximation

After layer normalization, high-dimensional visual tokens $\{\mathbf{m}_{v,j}\}$ are often nearly orthogonal [15, 21], implying $G_{ij} = \langle \mathbf{m}_{v,i}, \mathbf{m}_{v,j} \rangle \approx \gamma_j \delta_{ij}$, where $\gamma_j = \|\mathbf{m}_{v,j}\|_2^2$. Thus, $(G\boldsymbol{\alpha})_j \approx \gamma_j \alpha_j$. The exponent in Eq. (E.8) becomes $(2b_j - 2\gamma_j \alpha_j)/\tau$.

To achieve the common softmax form based on linear scores, we approximate by assuming the cross-interaction terms $b_j = \langle \boldsymbol{\mu}, \mathbf{m}_{v,j} \rangle$ dominate the self-interaction terms $2\gamma_j \alpha_j / \tau$ in determining the relative attention weights. This simplification, common in deriving attention mechanisms [41], effectively neglects the quadratic self-influence terms or treats their impact as uniform, yielding:

$$\alpha_j \approx \frac{\exp(2b_j/\tau)}{\sum_p \exp(2b_p/\tau)} = \operatorname{Softmax}_j\big(2\langle \boldsymbol{\mu}, \mathbf{m}_{v,j} \rangle / \tau\big). \tag{E.9}$$

This results in scores linear in $\boldsymbol{\mu}$ and $\mathbf{m}_{v,j}$ within the softmax, aligning with standard attention designs.

### E.1.5 Substituting a Geometric Surrogate for $\mu$

The inner products in Eq. (E.9) still involve the *unknown* mean $\mu$. We therefore introduce a linear surrogate driven by the *geometric cue* $\mathbf{M}_d$:

$$\mu \approx W_Q\mathbf{M}_d, \qquad \mathbf{m}_{v,j} \mapsto W_K\mathbf{m}_{v,j}, \tag{E.10}$$

a standard choice in multimodal transformers [47]. Choosing $\tau = \sqrt{d}$ converts Eq. (E.9) into the *scaled dot-product* of Vaswani et al. [48], and re-inserting the value tokens $\mathbf{m}_{v,j}$ finally yields Eq. (E.1).

### E.1.6 Consistency as $\tau \to 0$

Let $\boldsymbol{\alpha}_\tau^\star$ be the solution of the regularized optimization with temperature $\tau$. By the $\Gamma$-convergence of entropic regularization [9, Thm. 1], $\boldsymbol{\alpha}_\tau^\star \to \boldsymbol{\alpha}_0^\star$ as $\tau \to 0$, where $\boldsymbol{\alpha}_0^\star$ is the *exact* Euclidean projection solution of Eq. (E.2). Because $W_Q, W_K$ are trainable and $\mathbf{M}_d$ is fed through a temperature-scaling LayerNorm block, the network can approximate arbitrarily small $\tau$ in practice, so the learned weights converge to the optimal MMSE estimator.

### E.1.7 Connecting Back to Causal Front-Door

Finally, note that our estimator (E.1) uses *only* the bias-free query $\hat{\mathbf{M}}_d(x) \approx \mathbb{E}[\mathbf{M}_d \,|\, \mathcal{X} = x]$ and the raw visual tokens $\mathbf{M}_v(x)$. Because $\mathbf{M}_d \perp\!\!\!\perp \mathcal{U}$ by design, the attention weights $\boldsymbol{\alpha}(x)$ are conditionally independent of the confounder, guaranteeing that $\hat{\mathbf{M}}(x) = \text{Attn}(\hat{\mathbf{M}}_d, \mathbf{M}_v)$ is a *front-door-adjusted* proxy for the latent mediator, exactly as required by Perez et al. [41]. $\qquad\square$

## E.2 Derivation of the NWGM Approximation

This section provides a derivation for the Normalized Weighted Geometric Mean (NWGM) approximation used in main text Section 4.1 to implement the back-door adjustment for language de-biasing. The goal is to estimate the causal effect $P(\mathcal{Y} \,|\, \text{do}(\mathcal{T} = t))$.

The back-door adjustment formula (Eq. 1) is:

$$P(\mathcal{Y} \,|\, \text{do}(\mathcal{T} = t)) = \sum_z P(\mathcal{Y} \,|\, \mathcal{T} = t, \mathcal{Z} = z)P(\mathcal{Z} = z) = \mathbb{E}_{\mathcal{Z}}\big[P(\mathcal{Y} \,|\, t, z)\big]. \tag{E.11}$$

In a typical neural network classifier or segmenter, the conditional probability $P(\mathcal{Y} \,|\, t, z)$ is obtained by applying a $\text{Softmax}$ function to pre-activation scores (logits), denoted as $s_{\mathcal{Y}}(t, z)$. Thus, we are interested in computing:

$$P(\mathcal{Y} \,|\, \text{do}(\mathcal{T} = t)) = \mathbb{E}_{\mathcal{Z}}\big[\text{Softmax}(s_{\mathcal{Y}}(t, z))\big]. \tag{E.12}$$

The NWGM approximation (Eq. 2) states:

$$\mathbb{E}_{\mathcal{Z}}\big[\text{Softmax}\big(s_{\mathcal{Y}}(t, z)\big)\big] \approx \text{Softmax}\big(\mathbb{E}_{\mathcal{Z}}[s_{\mathcal{Y}}(t, z)]\big). \tag{E.13}$$

To justify this approximation, we first consider the relationship between the arithmetic mean and the weighted geometric mean (WGM). Let $f(\mathcal{Z})$ be a function of a random variable $\mathcal{Z}$ which takes values $z$ with probabilities $P(z)$. The arithmetic mean is:

$$\mathbb{E}_{\mathcal{Z}}[f(z)] = \sum_z f(z)P(z). \tag{E.14}$$

The weighted geometric mean is:

$$\text{WGM}_{\mathcal{Z}}[f(z)] = \prod_z f(z)^{P(z)}. \tag{E.15}$$

For many distributions, especially when the number of samples for $\mathcal{Z}$ is large or $f(z)$ does not vary excessively, the arithmetic mean can be approximated by the WGM:

$$\mathbb{E}_{\mathcal{Z}}[f(z)] \approx \text{WGM}_{\mathcal{Z}}[f(z)]. \tag{E.16}$$

Now, let $f(z) = \exp(g(z))$ for some function $g(z)$. Substituting this into the WGM definition (Eq. (E.15)):

$$\begin{aligned}
\text{WGM}_{\mathcal{Z}}[\exp(g(z))] &= \prod_z (\exp(g(z)))^{P(z)} \\
&= \prod_z \exp(g(z)P(z)) \\
&= \exp\left(\sum_z g(z)P(z)\right) \\
&= \exp(\mathbb{E}_{\mathcal{Z}}[g(z)]).
\end{aligned} \tag{E.17}$$

Combining Eq. (E.16) and Eq. (E.17):

$$\mathbb{E}_{\mathcal{Z}}[\exp(g(z))] \approx \exp(\mathbb{E}_{\mathcal{Z}}[g(z)]). \tag{E.18}$$

The $\mathrm{Softmax}$ function for a particular class output, given scores $s$, is proportional to $\exp(s_k)$ (where $s_k$ is the score for class $k$). Specifically, $\mathrm{Softmax}_k(s) = \frac{\exp(s_k)}{\sum_j \exp(s_j)}$. The NWGM approximation essentially applies the relationship in Eq. (E.18) to the logits *before* the normalization step inherent in Softmax, effectively moving the expectation inside the exponential terms that dominate the Softmax behavior. This leads to the approximation in Eq. (E.13):

$$\mathbb{E}_{\mathcal{Z}}\big[\,\mathrm{Softmax}\big(s_{\mathcal{Y}}(t,z)\big)\big] \approx \mathrm{Softmax}\big(\mathbb{E}_{\mathcal{Z}}[s_{\mathcal{Y}}(t,z)]\big). \tag{E.19}$$

This approximation is commonly used and has been discussed in works like Baldi and Sadowski [2] and Liu et al. [25].

With the additive score assumption (Section 4.1), $s_{\mathcal{Y}}(t,z) \simeq s_{\mathcal{T}}(t) + s_{\mathcal{Z}}(z)$, the expected score becomes:

$$\begin{aligned} \mathbb{E}_{\mathcal{Z}}[s_{\mathcal{Y}}(t,z)] &= \mathbb{E}_{\mathcal{Z}}[s_{\mathcal{T}}(t) + s_{\mathcal{Z}}(z)] \\ &= s_{\mathcal{T}}(t) + \mathbb{E}_{\mathcal{Z}}[s_{\mathcal{Z}}(z)]. \end{aligned} \tag{E.20}$$

Substituting this into Eq. (E.19) gives:

$$P(\mathcal{Y} \mid \mathrm{do}(\mathcal{T} = t)) \approx \mathrm{Softmax}(s_{\mathcal{T}}(t) + \mathbb{E}_{\mathcal{Z}}[s_{\mathcal{Z}}(z)]). \tag{E.21}$$

The de-confounded score used for prediction is therefore $s'_{\mathcal{Y}}(t) = s_{\mathcal{T}}(t) + \mathbb{E}_{\mathcal{Z}}[s_{\mathcal{Z}}(z)]$, as presented in main text Eq. 3. The term $\mathbb{E}_{\mathcal{Z}}[s_{\mathcal{Z}}(z)]$ is practically estimated by averaging the confounder embeddings $f_{\mathcal{Z}}(z_i)$ weighted by their empirical probabilities $P(z_i)$ from the training set, leading to the de-biased text representation $\mathbf{f}'_{\mathcal{T}}(t)$ in main text Eq. 4.

## E.3  Proof of Consistency of the Memory-Bank Estimator

In this appendix we provide a rigorous justification of the *Memory–bank estimator* defined in Eq. 9 of the main text. Recall that, for a fixed time index $t \in \mathbb{N}$, the estimator is

$$\hat{\mathbf{X}}_t = \sum_{\tau=1}^{W} w_{t,\tau}\,\mathbf{x}_{t-\tau}, \qquad w_{t,\tau} = \frac{\exp\big(\mathrm{sim}(\mathbf{x}_t, \mathbf{x}_{t-\tau})\big)}{\sum_{\sigma=1}^{W} \exp\big(\mathrm{sim}(\mathbf{x}_t, \mathbf{x}_{t-\sigma})\big)}. \tag{E.22}$$

Here $\mathbf{x}_t \in \mathbb{R}^d$ is the frame-level embedding for $x_t$, $W$ is the memory horizon, and $\mathrm{sim}(\cdot,\cdot)$ a bounded dot-product similarity.

Our goal is to show $\hat{\mathbf{X}}_t$ is a *consistent* estimator of the front–door expectation $\mathbb{E}_{\mathbf{X}'}[\mathbf{X}']$. The proof proceeds in two steps:

1. Show the *unweighted* empirical mean

$$\bar{\mathbf{X}}_t := \tfrac{1}{W} \sum_{\tau=1}^{W} \mathbf{X}_{t-\tau}$$

   is unbiased and converges a.s. to $\mathbb{E}_{\mathbf{X}'}[\mathbf{X}']$ under short–range stationarity.

2. Bound the bias introduced by the importance weights $\{w_{t,\tau}\}$ and prove it vanishes as $W \to \infty$.

### E.3.1  Preliminaries and Assumptions

**(A1) Short–range stationarity.**  Over the window of size $W$, the sequence $\{\mathbf{X}_{t-\tau}\}_{\tau=1}^{W}$ is wide–sense stationary, i.e. for any lag $\ell$, the joint law of $(\mathbf{X}_{t-\tau}, \mathbf{X}_{t-\tau-\ell})$ does not depend on $\tau$.

**(A2) Finite second moment.**  $\mathbb{E}\big[\|\mathbf{X}_0\|_2^2\big] < \infty$.

**(A3) Bounded similarity.**  There is $\kappa > 0$ such that $|\mathrm{sim}(\mathbf{x}, \mathbf{y})| \leq \kappa$ almost surely.

### E.3.2  Unweighted Empirical Mean

Under (A1), the past embeddings $\mathbf{X}_{t-1}, \ldots, \mathbf{X}_{t-W}$ form an i.i.d. sample from the marginal $P_{\mathbf{X}'}$. Hence

$$\mathbb{E}[\bar{\mathbf{X}}_t] = \mathbb{E}_{\mathbf{X}'}[\mathbf{X}'],$$

and by Kolmogorov's strong law of large numbers [16]

$$\bar{\mathbf{X}}_t \xrightarrow[W \to \infty]{\text{a.s.}} \mathbb{E}_{\mathbf{X}'}[\mathbf{X}']. \tag{E.23}$$

### E.3.3 Effect of Importance Weights

Write the weighted estimator as

$$\hat{\mathbf{X}}_t - \bar{\mathbf{X}}_t = \sum_{\tau=1}^{W} \left( w_{t,\tau} - \tfrac{1}{W} \right) \mathbf{X}_{t-\tau}.$$

Then

$$\|\hat{\mathbf{X}}_t - \bar{\mathbf{X}}_t\|_2 \;\leq\; \max_{\tau} \|\mathbf{X}_{t-\tau}\|_2 \left\| \mathbf{w}_t - \tfrac{1}{W}\mathbf{1} \right\|_1, \tag{E.24}$$

where $\mathbf{w}_t = (w_{t,1}, \ldots, w_{t,W})$. From (A3) the softmax weights satisfy

$$\frac{e^{-\kappa}}{We^{\kappa}} \;\leq\; w_{t,\tau} \;\leq\; \frac{e^{\kappa}}{We^{-\kappa}}, \quad \forall \tau,$$

so

$$\left\| \mathbf{w}_t - \tfrac{1}{W}\mathbf{1} \right\|_1 \;\leq\; 2\left(e^{2\kappa} - 1\right) W^{-1}. \tag{E.25}$$

Combining Eq. (E.24) with Eq. (E.25) and taking expectation gives

$$\mathbb{E}\left[ \|\hat{\mathbf{X}}_t - \bar{\mathbf{X}}_t\|_2 \right] \;\leq\; 2\left(e^{2\kappa} - 1\right) W^{-1} \, \mathbb{E}\left[ \max_{\tau} \|\mathbf{X}_{t-\tau}\|_2 \right], \tag{E.26}$$

where Doob's maximal inequality and (A2) ensure the last expectation is finite. Thus the weight-induced bias decays at rate $\mathcal{O}(W^{-1})$.

### E.3.4 Consistency

By the triangle inequality,

$$\|\hat{\mathbf{X}}_t - \mathbb{E}_{\mathbf{X}'}[\mathbf{X}']\|_2 \leq \|\hat{\mathbf{X}}_t - \bar{\mathbf{X}}_t\|_2 + \|\bar{\mathbf{X}}_t - \mathbb{E}_{\mathbf{X}'}[\mathbf{X}']\|_2.$$

The first term converges to 0 in $L^1$ by Eq. (E.26), the second converges a.s. by Eq. (E.23). Hence

$$\hat{\mathbf{X}}_t \;\xrightarrow[W\to\infty]{\text{prob.}}\; \mathbb{E}_{\mathbf{X}'}[\mathbf{X}'],$$

establishing consistency. $\qquad\square$

**Remark.** In practice $W$ is fixed (e.g. $W = 8$). Eq. (E.26) shows the residual bias is proportional to $(e^{2\kappa} - 1)/W$, and the model can trade off sharpness vs. bias via the learned scale $\kappa$.

### E.3.5 Necessity of Softmax Weights

In Eq. 9 of main paper the memory–bank feature is a convex combination $\hat{\mathbf{X}}_t = \sum_{\tau=1}^{W} w_\tau \mathbf{X}_{t-\tau}$, $\mathbf{w} \in \Delta^{W-1}$. Why choose the weights with a Softmax?

**Entropy–regularized MMSE derivation.** Let $\mathbf{z}^\star = \mathbb{E}[\mathbf{X}' \,|\, \mathbf{x}_t]$ be the ideal feature we would predict if the full distribution were known. Inside the attention–linear family we minimize the mean-square error while discouraging a single frame from monopolizing the weight:

$$\min_{\mathbf{w}\in\Delta^{W-1}} \left\| \sum_{\tau} w_\tau \mathbf{X}_{t-\tau} - \mathbf{z}^\star \right\|_2^2 \;-\; \lambda\, H(\mathbf{w}), \qquad H(\mathbf{w}) = -\sum_{\tau} w_\tau \log w_\tau, \tag{E.27}$$

where $H(\mathbf{w})$ is the Shannon entropy and $\lambda > 0$ is a temperature balancing *accuracy* and *diversity*. Introducing a Lagrange multiplier for $\sum_{\tau} w_\tau = 1$ and taking derivatives gives

$$w_\tau = \frac{\exp\left(\langle \mathbf{x}_t, \mathbf{x}_{t-\tau} \rangle / \lambda\right)}{\sum_{\sigma=1}^{W} \exp\left(\langle \mathbf{x}_t, \mathbf{x}_{t-\sigma} \rangle / \lambda\right)},$$

which is exactly the Softmax weight with scale $\lambda = \sqrt{d}/\kappa$ used in multi-head attention. Thus Softmax is the *unique* optimum of the entropy-regularized MMSE problem: it gives higher importance to frames more similar to the current one, yet avoids collapsing onto a single frame.

## F More Qualitative Analysis

This section provides further qualitative evidence to support the claims made in the main paper. We first delve into an analysis of the inherent biases present in the training dataset and then showcase additional comparative segmentation results that highlight the superior performance of our CERES framework.

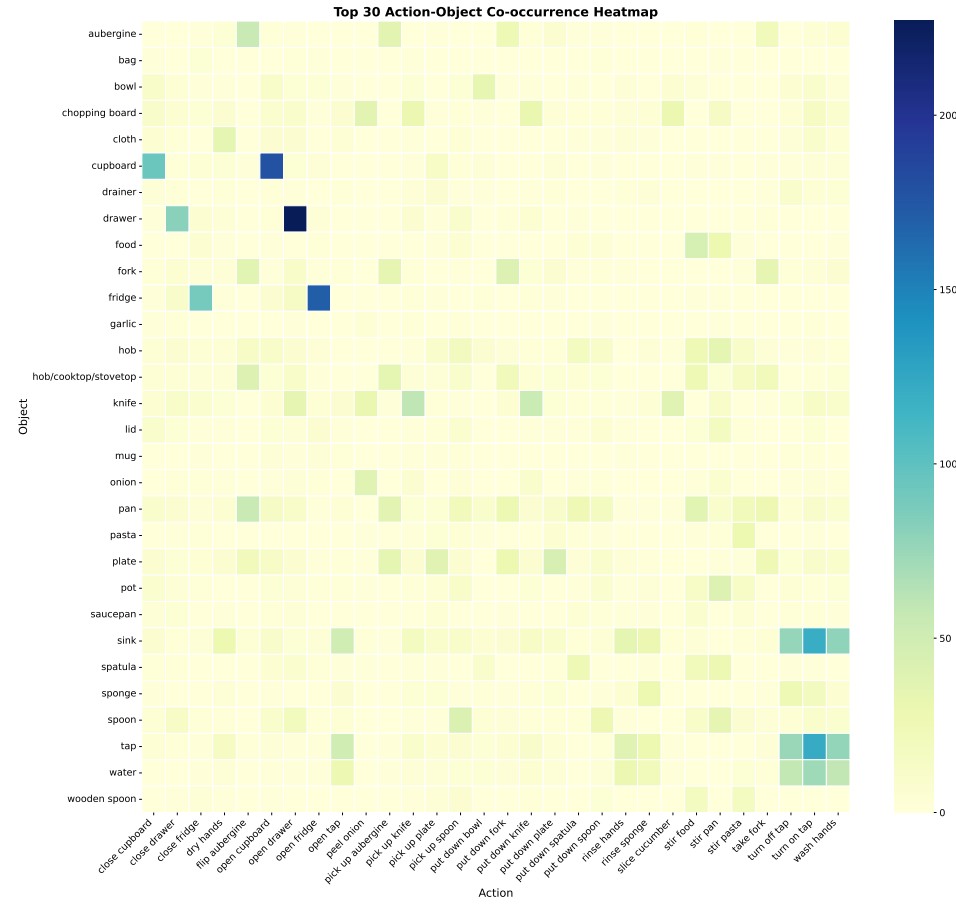

Figure F.1: Co-occurrence frequency of object categories and actions in the VISOR training set. Darker cells indicate more frequent pairings. This visualization clearly shows strong correlations between certain objects and actions, demonstrating the statistical bias our LBD module aims to mitigate.

## F.1 Analysis of Dataset Bias

As discussed in the main paper (Section 1), one of the primary challenges in Ego-RVOS is the presence of dataset biases, where certain object categories frequently co-occur with specific actions. This can lead models to learn spurious correlations rather than truly grounding the language query in the visual scene. To empirically demonstrate this, we conducted a statistical analysis of the action-object pairings within the VISOR training set.

Figure F.2 illustrates the distribution of the most frequent actions (verbs) in the training queries. It is evident that actions such as "cut", "take", and "put" are predominant.

Furthermore, Figure F.1 presents a heatmap (or a co-occurrence matrix analysis) showing the frequency of specific object categories (nouns) appearing with particular actions. This statistical imbalance naturally biases models trained on such data to favor common pairings, potentially failing on queries involving rarer but equally valid combinations.

These distributional statistics underscore the necessity for de-biasing mechanisms like the Linguistic Back-door Deconfounder (LBD) in CERES. By explicitly modeling and adjusting for these confounders (as defined by object-action pairs and their frequencies, see Section 4.1), CERES can achieve more robust language grounding and generalize better to less frequent or novel combinations.

## F.2 More Comparison of Segmentation Results

To further illustrate the robustness and accuracy of CERES, we present additional qualitative comparisons against the strong baseline, ActionVOS [36], on challenging sequences from the VISOR validation set. These examples complement those shown in Figure 4 in the main paper.

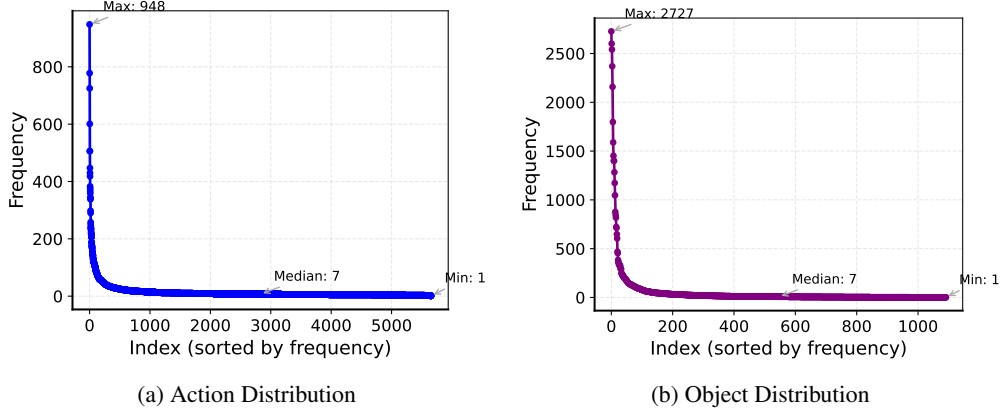

|                    | (a) Action Distribution | (b) Object Distribution |
|--------------------|-------------------------|-------------------------|

Figure F.2: Distribution of frequent actions and objects in the VISOR training set queries. This highlights a skewed distribution, where a few actions (objects) are significantly more common.

Table G.1: Performance (%) of CERES (ResNet101 backbone) on VISOR with different depth encoders for DAttn. Results are for the full CERES model. DAv2 refers to Depth Anything V2. The chosen configuration for our main experiments (DAv2 ViT-B) is highlighted. (**Bold** indicates the best performance; underlined indicates the second-best.)

| Depth Encoder | mIoU$^{\oplus}\uparrow$ | mIoU$^{\ominus}\downarrow$ | gIoU$\uparrow$ | Acc$\uparrow$ |
|---------------|------------------|------------------|-------|------|
| MiDaS (BEiT-L) | 60.8 | 15.0 | 70.6 | 73.7 |
| DAv2 (ViT-S) | 61.1 | **14.9** | 71.1 | 74.5 |
| DAv2 (ViT-B) | 64.0 | 15.3 | 72.4 | **76.3** |
| DAv2 (ViT-L) | **64.7** | 16.2 | **72.5** | 76.2 |

Figure F.3 showcases scenarios involving (1) significant hand-object occlusion, (2) rapid camera motion leading to motion blur, and (3) objects with subtle state changes.

These additional qualitative results, in conjunction with the quantitative improvements reported in the main paper, reinforce the conclusion that CERES's dual-modal causal intervention strategy effectively addresses key biases and confounding factors, leading to a more robust and reliable Ego-RVOS model. The VFD module enhances resilience to visual distortions common in egocentric video, while the LBD module improves generalization by mitigating reliance on spurious statistical correlations learned from biased training data.

# G   More Quantitative Analysis

This section provides additional quantitative analyses to further investigate the components and robustness of our CERES framework. All experiments are conducted on the VISOR dataset using the ResNet101 backbone, unless otherwise specified.

## G.1   Impact of Depth Encoder Choice in DAttn

The Visual Front-door Deconfounder (VFD) utilizes depth features to guide the aggregation of visual semantic features via Depth-guided Attention (DAttn). The quality and nature of these depth features can influence the effectiveness of the mediator construction. To assess this, we evaluated CERES with different pre-trained monocular depth estimation models as the source of depth features. We compared MiDaS (v3.1, BEiT-L backbone) [42] with various sizes of Depth Anything V2 (DAv2) [59] models, specifically those based on ViT-Small (vit-s), ViT-Base (vit-b), and ViT-Large (vit-l). The results are presented in Table G.1.

As shown in Table G.1, the choice of depth encoder impacts performance. Models from the Depth Anything V2 family generally outperform MiDaS in this application. Within the DAv2 family, there is a trend of improved performance with larger model sizes (ViT-S < ViT-B < ViT-L), with DAv2 (ViT-L) achieving the highest mIoU$^{\oplus}$ (64.7%) and gIoU (72.5%). However, DAv2 (ViT-B) provides a strong balance between performance (64.0% mIoU$^{\oplus}$, 72.4% gIoU) and computational cost/model size. Given this trade-off, we selected DAv2 (ViT-B) as the default depth encoder for CERES in our main experiments, as its results are highly competitive while being more resource-efficient than the ViT-L variant. The slightly higher mIoU$^{\ominus}$ for larger models might indicate a

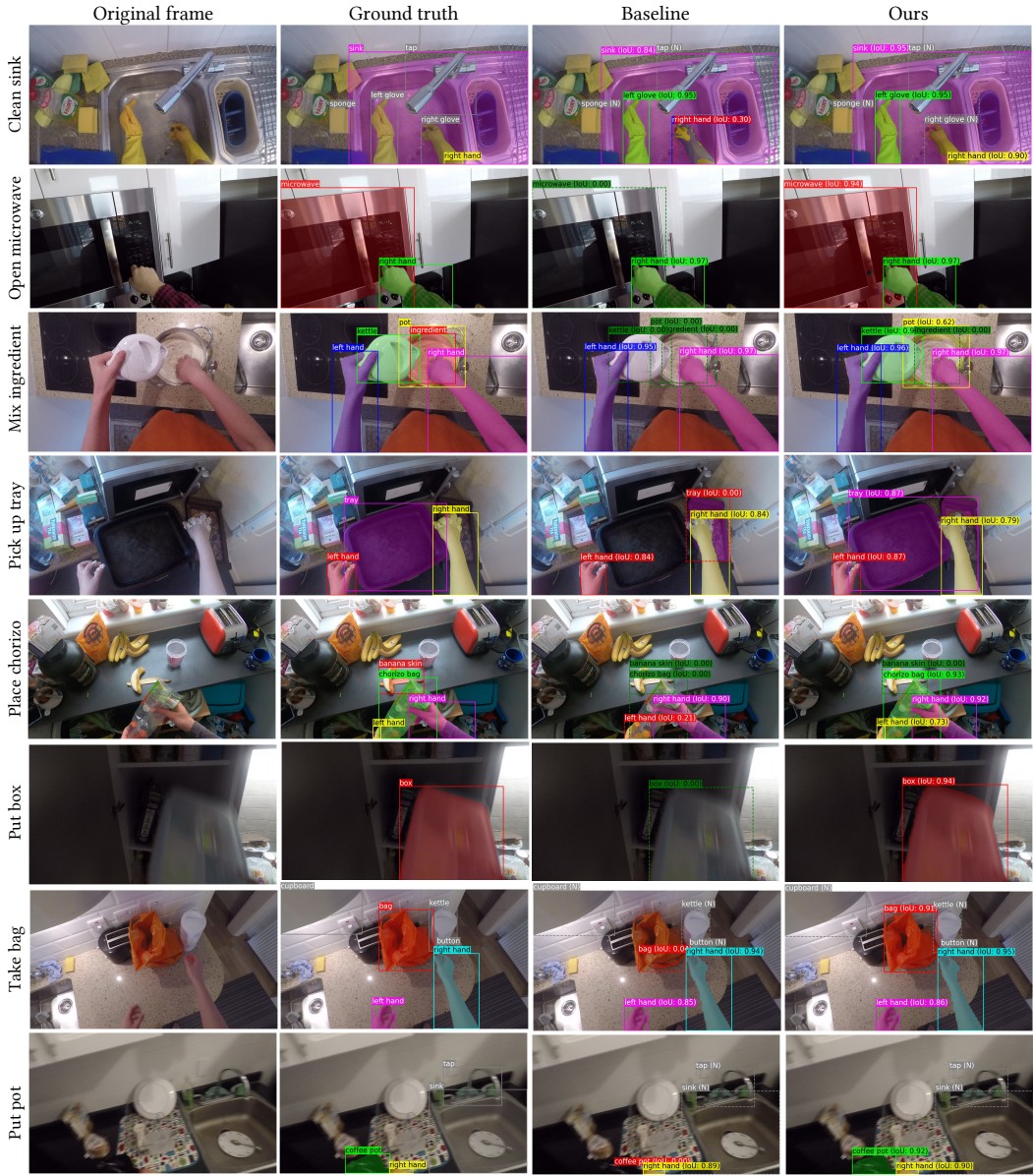

Figure F.3: Additional qualitative comparisons between ActionVOS (Baseline) and CERES on challenging Ego-RVOS scenarios. CERES consistently demonstrates more robust and accurate segmentation in the presence of occlusions, motion blur, and subtle interactions.

more complex feature space that could require further fine-tuning or regularization if negative object suppression is a primary concern. Overall, these results confirm that higher-quality depth information, as provided by more recent and powerful depth estimation models, contributes positively to the VFD module's ability to de-bias visual features.

## G.2 Robustness to Depth Feature Degradation

To further assess the robustness of our DAttn mechanism to imperfections in depth information, we simulated scenarios where depth map quality might be compromised (e.g., due to challenging scenes, sensor noise, or limitations of the depth estimator). We conducted an experiment by adding varying levels of Gaussian noise to the normalized depth features extracted by the DAv2 (ViT-B) encoder before they are fed into the DAttn module. Specifically, for each depth feature vector $f_d$, we added noise $\epsilon \sim \mathcal{N}(0, \sigma_n^2 I)$, where $\sigma_n$ is the standard

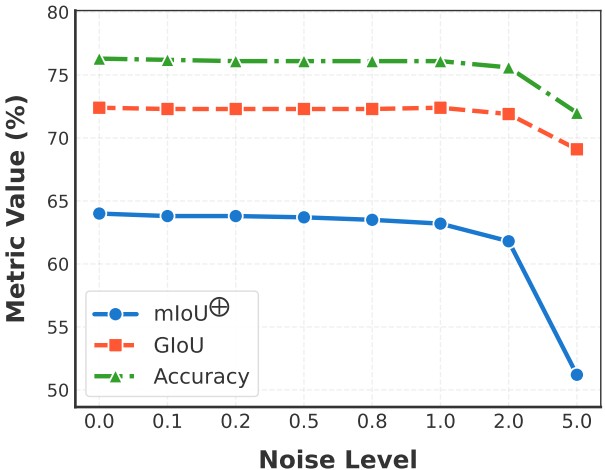

Figure G.1: Performance of CERES (ResNet101, DAv2 ViT-B depth) on VISOR under varying levels of Gaussian noise added to depth features. mIoU$^\oplus$ (blue), gIoU (orange), and Accuracy (green) are plotted against the noise standard deviation ($\sigma_n$). ($\sigma_n = 0.0$ means no noise.)

Table G.2: Computational overhead at $448 \times 448$ on RTX 3090. (**Bold** indicates the best; underlined indicates the second-best. Params include the frozen depth encoder (DAv2-B); FPS uses a memory window $W{=}5$.)

| Method | Backbone | Params (M)↓ | FPS↑ | mIoU$^\oplus$↑ |
|---|---|---|---|---|
| ActionVOS | ResNet101 | **195** | **23.8** | 58.4 |
| ActionVOS | VSwin-B | 237 | 15.4 | 62.9 |
| CERES (Ours) | ResNet101 + DAv2-B | 306 | 18.2 | **64.0** |

deviation of the noise. The performance of CERES on VISOR was evaluated across a range of noise levels $\sigma_n \in [0.0, 0.1, 0.2, 0.5, 0.8, 1.0, 2.0, 5.0]$.

The results, depicted in Figure G.1, show a clear trend. For low to moderate noise levels, CERES exhibits notable resilience. Specifically, when the noise standard deviation $\sigma_n$ is less than 1.0 (i.e., for $\sigma_n \leq 0.8$ in our tested discrete levels), the impact on performance is minimal. At $\sigma_n = 0.8$, mIoU$^\oplus$ drops from 64.0% to 63.5% (a relative decrease of 0.78%), gIoU drops from 72.4% to 72.3% (a relative decrease of 0.14%), and Accuracy drops from 76.3% to 76.1% (a relative decrease of 0.26%). In all these cases, the relative performance degradation is less than 1% compared to the no-noise baseline. Even at $\sigma_n = 1.0$, gIoU remains remarkably stable (72.4%) and Accuracy only slightly decreases to 76.1%, while mIoU$^\oplus$ sees a modest drop to 63.2% (a 1.25% relative decrease).

As the noise intensity increases further (e.g., $\sigma_n = 2.0$ and $\sigma_n = 5.0$), the performance degradation becomes more pronounced, particularly for mIoU$^\oplus$, which drops to 61.8% and 51.2% respectively. This indicates that while DAttn can effectively handle minor inaccuracies in depth features, highly corrupted geometric information will naturally lead to a more significant decline in segmentation quality.

This graceful degradation under low to moderate noise levels (with performance loss under 1% for $\sigma_n < 1.0$) demonstrates the robustness of our causally-inspired vision-depth fusion approach. The DAttn mechanism appears capable of leveraging the general structure provided by depth cues even when they are not perfectly accurate, while still underscoring the overall benefit of reasonably high-quality depth information for optimal performance.

## G.3 Overhead Comparison

To quantify the computational overhead introduced by our causal framework relative to established baselines, we report end-to-end throughput and parameter counts at $448 \times 448$ input resolution measured on an NVIDIA RTX 3090. The table below reproduces the setup described in the rebuttal and serves as the basis for overhead comparison.

According to Table G.2, relative to the ResNet101 ActionVOS baseline, our method increases the parameter count by approximately $+56.9\%$ (from 195M to 306M) and reduces throughput by $-23.5\%$ (from 23.8 to 18.2 FPS), reflecting the added depth pathway. In contrast, when compared to the stronger VSwin-B ActionVOS baseline, our method carries a smaller parameter overhead of $+29.1\%$ (from 237M to 306M) yet delivers higher throughput by $+18.2\%$ (from 15.4 to 18.2 FPS), indicating that the observed efficiency gains stem from the causal design rather than scaling the RGB backbone alone.

Overall, these measurements isolate the overhead attributable to the depth encoder and the lightweight attention modules. While the separate depth pathway increases parameters and reduces FPS versus a ResNet101-only baseline, the framework remains more efficient than upgrading the RGB backbone to Video Swin-B, achieving higher throughput at comparable or better accuracy.

