# OpenReview forum: "Robust Egocentric Referring Video Object Segmentation via Dual-Modal Causal Intervention"
_NeurIPS.cc/2025/Conference — NeurIPS 2025 poster_

### Official Review · Reviewer_VF42 · 2025-06-29

**Clarity:** 4
**Significance:** 3
**Originality:** 3
**Rating:** 4
**Confidence:** 2

**Summary:**

This paper discusses Egocentric Referring Video Object Segmentation (Ego-RVOS), a task focused on segmenting objects involved in human actions in egocentric vision, guided by language queries. The task is challenging due to language biases and limitations of the egocentric view. In this work, the authors introduce Causal Ego-Referring Segmentation (CERES), a novel framework leveraging causal inference. CERES applies principles based on back door adjustment and front door adjustment to counteract the confounding issues. The proposed model was evaluated on three public egocentric vision datasets and achieved state-of-the-art performance.

**Questions:**

1. In table 1, why is "Ours" in bold but have higher mIoU⊖ and cIoU⊖?
2. In table 2, ReferFormer+ the second best mIoU⊖, why is "ours" underlined?
3. In table 4, adding LBD shows clear advantage when not using DAttn and MAttn. However, the difference in numbers between the last two rows (w/ and w/o LBD) is small. Are there more analyses that justify the effectiveness of LBD in the final design?
4. This work identified two confounding sources: observable language bias and unobservable visual confounding. Is there existing work or analysis supporting these as the primary confounding factors?

**Ethical Concerns:**

["NO or VERY MINOR ethics concerns only"]

**Final Justification:**

The rebuttal answered all my questions and I will maintain my original rating.

**Limitations:**

Yes

**Quality:**

3

**Strengths And Weaknesses:**

Strength:
1. The clarity of this paper is very good. Overall, this paper is well-written, with clear format, tables and math.
2. The proposed method is evaluated across multiple datasets, achieving state-of-the-art performance.
3. The ablation study results reported in this work support the framework design choices and helps understanding.

Major weakness:

N/A

Minor weakness:

Line 31: "because because".

Some tables (1,2,4) may be inaccurate and requires clarification (see questions).

Over all, this paper is well-written. It introduces an interesting idea and the experimental results show potential . Due to limited understanding behind the math in causal reasoning, my current rating for this paper is borderline accept, and will be adjusted after rebuttal.

---

> ### Author Rebuttal · Authors · 2025-07-31
>
> We sincerely thank you for your valuable time and the positive and constructive assessment. We are delighted that you found our paper to be **well-written and clear**, our results to achieve **state-of-the-art performance** across multiple datasets, and our ablation studies to be **supportive and insightful**.
>
> Crucially, we appreciate that you recognized the core contribution of our work as a **"novel framework leveraging causal inference"** to tackle the complex challenges of Egocentric Referring Video Object Segmentation. Your thoughtful questions have allowed us to further clarify the foundations and impact of our method, and we have addressed each point in detail below.
>
> ---
>
> > `Minor weakness`
>
> We are sincerely grateful for your meticulous review and conscientious attention to detail. Your careful and responsible reading has helped us identify and correct several minor yet important errors, significantly improving the quality and precision of our paper.
>
> > `Weakness 1 / W1` Line 31: "because because".
> >
> > `Question 1 / Q1` In table 1, why is "Ours" in bold but have higher mIoU⊖ and cIoU⊖?
> >
> > `Q2` In table 2, ReferFormer+ the second best mIoU⊖, why is "ours" underlined?
>
> You are absolutely correct on all three points you raised:
> - The typo on Line 31 ("because because") was an oversight.
> - The bolding in **Table 1** for `mIoU⊖` and `cIoU⊖` was incorrect. As lower is better, ReferFormer+ indeed achieves the best scores for these metrics with the R101 backbone.
> - Similarly, the underlining in **Table 2** for `mIoU⊖` was a mistake. HOS and ReferFormer+ have the first and second-best scores, respectively.
>
> Our main performance claims are based on the primary metrics (mIoU↑, gIoU↑, etc.), where our method does achieve state-of-the-art results. We have corrected the bolding and underlining in the final version to accurately reflect the top performers for every metric. We are very grateful for this careful correction.
>
>
> ---
>
> > `Q3` In table 4, adding LBD shows clear advantage when not using DAttn and MAttn. However, the difference in numbers between the last two rows (w/ and w/o LBD) is small. Are there more analyses that justify the effectiveness of LBD in the final design?
>
> Thank you for this insightful question.
>
> The primary role of Linguistic Back-door Deconfounder (LBD) is to mitigate the model's over-reliance on spurious language-level correlations, a bias that is most detrimental when encountering rare object-action pairs. To isolate and highlight this specific contribution, we performed a new ablation study on the VISOR "hard" subset, which is composed of such rare concepts. The results below demonstrate that LBD is, in fact, the most impactful component for improving generalization in these challenging cases.
>
> Table: Ablation on the VISOR "hard" subset.
> |Method|IoU⊕|gIoU|
> |-|-|-|
> |ActionVOS|58.40|69.90|
> |+ LBD only|61.4 (+3.0)|71.8 (+1.9)|
> |+ VFD only|60.5 (+2.1)|71.3 (+1.4)|
> |CERES|62.3 (+3.9)|72.2 (+2.3)|
>
> As shown, adding LBD alone provides a larger gain in mIoU⊕ (+3.0%) than adding the VFD alone (+2.1%) on this difficult subset. This directly confirms that LBD is highly effective at addressing its targeted bias, forcing the model to ground its decisions in visual evidence rather than statistical shortcuts. While the Visual Front-door Deconfounder (VFD) provides a significant general performance boost, the LBD is a crucial component for robust generalization.
>
> We will add this new table and analysis to the paper to better clarify the distinct and critical role of LBD.
>
> ---
> > `Q4` This work identified two confounding sources: observable language bias and unobservable visual confounding. Is there existing work or analysis supporting these as the primary confounding factors?
>
> Thank you for this important question, which allows us to clarify the established foundation of our work.
>
> Our selection of confounders is a direct response to two distinct challenges well-documented in vision-language research. Specifically, we address (1) **observable language biases arising from dataset statistics**, a known problem tackled with backdoor adjustment in prior work [1, 2], and (2) **unobservable visual confounding from inherent egocentric distortions** like occlusion and motion blur, which have been identified as key challenges in egocentric video analysis [3]. Our unified framework, which applies the appropriate causal tool to each problem, draws inspiration from comprehensive causal approaches [4]. Applying a formal causal model to diagnose and address these specific challenges in the context of egocentric video is a key innovation of our work.
>
> Below, we elaborate on these two pillars of our methodology:
>
> 1. **Observable Language Bias (Z)**:
>
> The problem of models exploiting statistical shortcuts instead of learning genuine relationships is a known source of failure. In the image captioning domain, Liu et al. (2022) [1] explicitly identify and address a "linguistic confounder" using backdoor adjustment to prevent the model from relying on spurious object-attribute correlations. We argue that **Ego-RVOS faces an analogous challenge**, where frequent verb-noun pairings in the training data (e.g., "cut" consistently appearing with "knife") create a powerful but brittle shortcut. We model this as an observable confounder Z, as its statistics are directly computable from the dataset. Applying backdoor adjustment is, therefore, a principled and established strategy to block this spurious causal path.
>
> 2. **Unobservable Visual Confounding (U):**
>
> The inherent nature of egocentric video introduces a separate class of confounders. As Shen et al. [3] and others have noted, factors like severe **"occlusion," "blur," and "frequently changing object appearance"** are defining characteristics of egocentric data. These visual artifacts are not explicitly labeled, yet they directly impact both the visual input X and the final segmentation Y. This makes them a classic unobserved confounder U, for which backdoor adjustment is impossible. This motivates our use of the front-door criterion, and our primary innovation here is the design of a novel vision-depth mediator specifically engineered to be robust to these distortions, thereby isolating the true causal effect of the visual input.
>
> In the final version of our paper, we will revise the related work section to incorporate this discussion and these citations more explicitly, making it clear how our work builds upon these foundational insights to address the specific, dual challenges of robust Ego-RVOS.
>
>
> [1] Bing Liu, et al. Show, deconfound and tell: Image captioning with causal inference. In CVPR, 2022.
>
> [2] Xu Yang, et al. Causal Attention for Vision-Language Tasks. In CVPR, 2021.
>
> [3] Yuhan Shen, et al. Learning to segment referred objects from narrated egocentric videos. In CVPR, 2024.
>
> [4] Liuyu Wang, et al. Vision-and-language navigation via causal learning. In CVPR, 2024.

---

> > ### Comment · Reviewer_VF42 · 2025-08-05
> >
> > Thank you for the additional explanation of the methodology and agreeing to correct the errors.

---

### Official Review · Reviewer_gqxt · 2025-06-29

**Clarity:** 2
**Significance:** 2
**Originality:** 2
**Rating:** 4
**Confidence:** 3

**Summary:**

This paper proposes a new framework, CERES, for Egocentric Referring Video Object Segmentation (Ego-RVOS). The core idea is to improve model robustness by addressing two sources of spurious correlations using principles from causal inference.

First, to counteract language biases in datasets (e.g., "knife" always co-occurring with "cut"), the framework uses back-door adjustment on the text features, which fix the bias using dataset statistics.
Second, to handle visual confounding factors common in egocentric video (e.g., motion blur, occlusions), it applies front-door adjustment. A key component of this is a novel visual mediator that uses geometric depth features, which are assumed to be more robust to confounders, to guide the aggregation of semantic RGB features via an attention mechanism.

The authors demonstrate that CERES achieves state-of-the-art performance on several Ego-RVOS benchmarks.

**Questions:**

My major concern is that this paper should have been built upon a comprehensive set of suitable baselines due to its heavy reliance on the pretrained models, followed by incremental experiments incorporating +CERES to demonstrate its generalizability, rather than being presented as a standalone SOTA method for comparison.
For example, this paper demonstrates the feasibility on ActionVOS, but does not provide evidence of its applicability to other pre-trained models, such as RefFormer or HOS.

**Ethical Concerns:**

["NO or VERY MINOR ethics concerns only"]

**Final Justification:**

Thanks the authors for their detailed responses, which partly resolves my major concern stated in “Questions”. I decide to raise my score to “borderline accept”.

**Limitations:**

yes

**Quality:**

2

**Strengths And Weaknesses:**

Strengths:
- The performance of this paper is good, showing significant improvement over the baseline ActionVOS.
- The method used in this paper is effective, as it takes into account the bias in text-based datasets and integrates depth information visually.


Weakness：
- This paper lacks a well-defined motivation for incorporating depth information, which provides the largest improvement, and makes an unfair comparison with the baselines. In Table 4, it can be observed that the introduction of depth information through DAttn yields the most significant improvement, with the mIoU increasing from 59.9 in the first row to 63.3 in the third row. Moreover, it is widely acknowledged that depth information provides a strong prior for segmentation, as evidenced both intuitively and through experiments such as those in DepthAnything. Therefore, I think the way this paper addressing motion blur is somewhat trivial, and I remain skeptical of the theoretical framework presented in this paper.
- Using a pre-trained model instead of training from scratch reduces the originality of this paper. Line 269 mentions the use of a pre-trained model, and the number in the first row of Table 4 is identical to that of the ActionVOS method with a Res101 backbone in Table 1. This indicates that the pre-trained model from ActionVOS was used. In other words, the method's performance improvement is built upon the combination of a pre-trained model and depth information, which, to some extent, affects the validity of this paper.
- This illustrations, especially the egocentric images, are very difficult to see clearly, including Figure 1, Figure 3, and Figure 4.

---

> ### Author Rebuttal · Authors · 2025-07-31
>
> We sincerely thank you for your detailed feedback and for recognizing the key strengths of our work. We are pleased that you found our causally-motivated approach **effective** and our results **good**, noting the **significant improvement** over the baseline. We are particularly encouraged that you acknowledged the core of our contribution: a principled method that corrects for text-based dataset bias and visually integrates depth information via a **novel visual mediator**.
>
> In our response, we will address your valid concerns regarding the motivation for depth `Weakness 1 / W1`, the fairness of our comparisons `W1`, and the generalizability of our framework `W2, Question 1 / Q1`.
>
> ---
>
> > `W1` This paper lacks a well-defined motivation for incorporating depth information, which provides the largest improvement, and makes an unfair comparison with the baselines. In Table 4, it can be observed that the introduction of depth information through DAttn yields the most significant improvement, with the mIoU increasing from 59.9 in the first row to 63.3 in the third row. Moreover, it is widely acknowledged that depth information provides a strong prior for segmentation, as evidenced both intuitively and through experiments such as those in DepthAnything. Therefore, I think the way this paper addressing motion blur is somewhat trivial, and I remain skeptical of the theoretical framework presented in this paper.
>
> We sincerely thank you for your thoughtful and challenging questions. These points are crucial, and we appreciate the opportunity to clarify the motivation for our work, the fairness of our evaluation, and the novelty of our theoretical framework.
>
> 1. **Causal Motivation for Integrating Depth**.
>
> We agree that depth is a strong prior for segmentation. However, our motivation is not simply to leverage this prior for a performance boost. Instead, our core contribution is a **causal framework** designed to mitigate specific confounding effects in egocentric video, and depth information is the chosen instrument to realize our proposed causal intervention.
>
> Our central hypothesis is this: unobserved confounders ($U$) like **occlusion and motion blur** primarily corrupt the semantic information available in RGB frames ($X$), leading to biased features. However, the scene's underlying geometric structure, captured by a depth map, is significantly more robust to these specific visual distortions. For example, when a hand occludes a knife, the knife's semantic RGB features become entangled with the hand's, but the depth map preserves their distinct spatial planes.
>
> Theoretically, we frame this using **front-door adjustment**. Our Depth-guided Attention (DAttn) module is the explicit mechanism for this intervention. It uses the robust geometric mediator ($M_d$ from depth) to guide and disambiguate the less reliable semantic mediator ($M_v$ from RGB). This principled design directly blocks the confounding path from the visual input to the output, making it far more than a trivial application of a known prior.
>
> 2. **On the Fairness of Comparison.**
>
> To demonstrate that our contribution lies in how depth is integrated, not merely its presence, we respectfully point to the ablation study already in our paper.
>
> - Evidence from Table 4: We have explicitly compared our DAttn against **a naive MLP-based RGB+depth fusion (Row 3 vs. Row 4 in Table 4)**. The results are telling:
>   - The simple MLP fusion improves target segmentation (mIoU⊕ from 59.9% to 62.1%) but **worsens discrimination**, increasing errors on non-target objects (mIoU⊖ from 16.3 to 17.5).
>   - In contrast, our causally-motivated DAttn not only achieves superior target segmentation (mIoU⊕ 63.3%) but also improves discrimination, reducing non-target errors (mIoU⊖ to 15.8).
>
> This directly validates that the structure of our intervention is **critical and non-trivial**. The comparison with baselines is therefore a fair evaluation of our causal framework against non-causal alternatives. The performance gain is evidence that our method for addressing confounding is effective.
>
> 3. **New Supporting Evidence for our Theoretical Framework.**
>
> To provide more direct, intuitive evidence for our claims, we have produced new attention visualizations for challenging occlusion scenarios. We observe that:
> - In a representative case, the baseline model's attention incorrectly shifts to the occluding "hand".
> - In contrast, our model, guided by DAttn, maintains a tight and accurate focus on the occluded "knife".
>
> We will add these compelling visualizations to the appendix.
>
> ---
>
> > `W2` Using a pre-trained model instead of training from scratch reduces the originality of this paper. Line 269 mentions the use of a pre-trained model, and the number in the first row of Table 4 is identical to that of the ActionVOS method with a Res101 backbone in Table 1. This indicates that the pre-trained model from ActionVOS was used. In other words, the method's performance improvement is built upon the combination of a pre-trained model and depth information, which, to some extent, affects the validity of this paper.
>
> > `Q1` My major concern is that this paper should have been built upon a comprehensive set of suitable baselines due to its heavy reliance on the pretrained models, followed by incremental experiments incorporating +CERES to demonstrate its generalizability, rather than being presented as a standalone SOTA method for comparison. For example, this paper demonstrates the feasibility on ActionVOS, but does not provide evidence of its applicability to other pre-trained models, such as RefFormer or HOS.
>
>
> We appreciate your insightful feedback and appreciate the opportunity to clarify our contribution.
>
> First, we would like to precisely clarify our model's heritage, as our method builds upon the **pre-trained weights of ReferFormer, not those of ActionVOS**. To be more specific, we inherited the successful **task-specific framework** proposed by ActionVOS for Ego-RVOS, which includes their designs for using an "action as prompt", a "positive prediction head", and an "Action-guided Focal Loss".
>
> The identical performance between our baseline (Table 4, row 1) and the ActionVOS results (Table 1) is intentional and serves to demonstrate that we have faithfully reimplemented their framework as a strong starting point.
>
> However, our key observation, which motivates this work, is that these foundational models are often **pre-trained on large-scale third-person video datasets** and ***do not*** **explicitly account for the unique biases inherent in the egocentric perspective**. Our core innovation then builds upon this baseline by introducing a principled causal framework (CERES) specifically designed to address the unique visual and linguistic biases in egocentric video that are overlooked by standard models.
>
> To directly address your valid concern about generalizability, we have conducted new experiments showing our framework's effectiveness on the original ReferFormer model, further demonstrating its modularity. We agree completely with your excellent suggestion that framing our work as a generalizable framework is a more powerful way to convey its value. The results below confirm that our modules provide a notable performance lift on a different architecture, proving they are ***not*** **overfitted to the ActionVOS design**:
>
> Table: Applying CERES to ReferFormer (ResNet101) on VISOR
> |Method|mIoU⊕ (%) ↑|gIoU (%) ↑|
> |-|-|-|
> |ReferFormer|59.9|63.1|
> |+CERES (Ours)|62.5 (+2.6)|66.8 (+3.7)|
>
> These positive results encourage us to view CERES as a **foundational approach for improving robustness** across various egocentric vision tasks. We believe the same causal principles could be highly effective for other challenges like egocentric referring expression comprehension or video question answering.
>
> In summary, we thank you for your valuable guidance. We will revise the manuscript to better articulate these details and frame our contributions as a generalizable causal framework, and we look forward to exploring these broader applications in future work.
>
> ---
>
> > `W3` These illustrations, especially the egocentric images, are very difficult to see clearly, including Figure 1, Figure 3, and Figure 4.
>
> Thank you for your valuable feedback regarding the clarity of our figures. These images indeed need to be enlarged to be clear.
>
> To rectify this, we will add high-resolution, enlarged versions of these figures to the appendix in our revised manuscript. We will also improve their size and layout within the main text for the camera-ready version as space allows.

---

> > ### Comment · Reviewer_gqxt · 2025-08-05
> > **Response to authors’ rebuttal**
> >
> > Thanks the authors for their detailed responses, which partly resolves my major concern stated in “Questions”. I decide to raise my score to “borderline accept”.

---

### Official Review · Reviewer_myER · 2025-07-01

**Clarity:** 3
**Significance:** 3
**Originality:** 2
**Rating:** 4
**Confidence:** 4

**Summary:**

This paper proposes a new RVOS method, named CERES. CERES addresses two challenges: confounders in visual images and language representation biases, respectively. Specifically, the visual confounders are addressed by intergrating depth data, while the language presentation biases is addressed by intergrating data statistics.

**Questions:**

Please refer to weaknesses.

**Ethical Concerns:**

["NO or VERY MINOR ethics concerns only"]

**Final Justification:**

Thanks for the authors' reply and extra analysis of my questions. Now I have a clearer understanding of CERES's ability in state-change scenarios and open-vocabulary settings. I have decided to raise my rating.

**Quality:**

2

**Strengths And Weaknesses:**

Strengths:
1. CERES achieves state-of-the-art results and significantly outperforms ActionVOS, demonstrating its strong effectiveness.
2. The consideration of biases in both visual and textual information is novel and interesting.
3. CERES performs well across different network backbones, highlighting its practical applicability and flexibility.

Weaknesses:
1. The integration of data statistics and depth information seems somewhat detached from the earlier causal bias analysis. The method mainly improves performance by adding depth data and data statistics to the RVOS model through mechanisms like attention or concatenation, but the connection to the prior bias framework is not clearly established.
2. While ActionVOS includes analysis of state-changing objects, it remains unclear whether CERES can be effectively applied to such scenarios.
3. It would be valuable to understand how CERES performs in open-vocabulary settings, particularly on unseen objects or actions.

---

> ### Author Rebuttal · Authors · 2025-07-31
>
> We sincerely thank you for your thorough evaluation and constructive feedback. We are encouraged that you found our work to be effective, novel, and practical, highlighting that CERES achieves **state-of-the-art results**, that our **dual-modal causal approach is novel and interesting**, and that its **strong performance across different backbones shows its practical applicability**.
>
>
> Your main suggestions provide an excellent opportunity for us to clarify key aspects of our method. Below, we address each point in detail, using existing experimental evidence to: (1) further clarify the principled connection between our causal theory and its implementation `Weakness 1 / W1`; (2) demonstrate CERES's effectiveness on state-changing objects `W2`; and (3) highlight its strong open-vocabulary performance on unseen concepts `W3`.
>
> ---
>
> > `W1` The integration of data statistics and depth information seems somewhat detached from the earlier causal bias analysis. The method mainly improves performance by adding depth data and data statistics to the RVOS model through mechanisms like attention or concatenation, but the connection to the prior bias framework is not clearly established.
>
> We sincerely thank you for this insightful comment. It highlights a crucial point: the connection between our causal framework and its practical implementation must be explicit and well-justified. We agree that this link is the most important part of our paper, and we appreciate the opportunity to clarify it.
>
> Our architectural choices are not ad-hoc additions but are, in fact, principled and direct implementations of the causal adjustment formulas derived from our analysis. The key contribution is ***how*** **we handle linguistic bias and visual bias, which is dictated entirely by our causal graph (Fig. 1c)** to solve two distinct problems.
>
> 1. **Linguistic Bias → Back-door Adjustment (Using Data Statistics):**
>    - **The Problem:** Models learn dataset shortcuts, like assuming the query "cut carrot" means *any* knife in the scene should be segmented, even one lying inactive on the counter. This is an observable confounding bias `Z` linking text `T` to output `Y`.
>    - **The Causal Solution:** To break this shortcut, we apply the back-door adjustment formula, $\mathbb{E}_z[P(Y|T,z)]$.
>    - **Our Principled Implementation (LBD):** Our LBD module (Eq. 4) implements this by calculating an "average confounding effect" from the dataset statistics and using it to de-bias the text features. This is not arbitrary concatenation; it is a specific operation designed to remove the statistical shortcut. This forces the model to ground the language query in *visual evidence of the action*, rather than just co-occurrence statistics.
> 2.  **Visual Bias → Front-door Adjustment (Using Depth-guided Attention):**
>     - **The Problem:** Egocentric video is plagued by occlusions and motion blur. This is an **unobserved** confounder `U` that corrupts the visual input `X`. We cannot adjust for `U` directly.
>     - **The Causal Solution:** We apply the front-door adjustment, which requires a mediator `M` that is robust to the confounder `U`.
>     - **Our Principled Implementation (DAttn):** A standard visual feature `M_v` (from RGB) is confounded by occlusion. However, an object's geometric structure `M_d` (from depth) remains far more stable. Our Depth-guided Attention (DAttn) is the direct embodiment of this principle: we use the **robust depth features (`M_d`) as the Query to guide attention over the confounded RGB features (`M_v` as Key/Value)**. Intuitively, the model is forced to ask using stable geometry: *"Where in this occluded, blurry visual scene is the object that has this specific shape and spatial location?"*
>
> To provide empirical evidence that this principled design is superior to simple feature fusion, our ablation in **Table 4** shows that our causally-guided DAttn (**63.3% mIoU⊕**) substantially outperforms a naive MLP-based depth fusion (**62.1% mIoU⊕**). This result confirms that *how* depth is integrated, guided by the front-door principle, is more critical than its mere presence.
>
> We sincerely thank you again for this valuable feedback. In our final version, we will revise the methodology section to more explicitly and forcefully articulate these connections, ensuring the link between our causal theory and each architectural choice is unmistakable.
>
> ---
> > `W2` While ActionVOS includes analysis of state-changing objects, it remains unclear whether CERES can be effectively applied to such scenarios.
>
> Thank you for this insightful question regarding the model's performance on state-changing objects. We agree that this is a critical aspect of robust video object segmentation.
>
> To address this, we have evaluated our method on the VSCOS dataset, which is specifically **designed to assess performance on objects undergoing significant transformations**. Our results, presented in Table 3, show that CERES achieves a higher **mIoU of 55.3% compared to 52.5%** for ActionVOS on this dataset.
>
> To further highlight this capability, we will add qualitative visualizations from the VSCOS dataset to the appendix, showcasing successful segmentation of objects during state changes.
>
> ---
> > `W3` It would be valuable to understand how CERES performs in open-vocabulary settings, particularly on unseen objects or actions.
>
> Thank you for this insightful question regarding the model's open-vocabulary performance. We agree that evaluating on unseen concepts is crucial for demonstrating robustness.
>
> We addressed this in two ways in our experiments, which we will clarify further in the final version:
> 1. **Unseen Concepts within VISOR (Table 2):** Thank you for highlighting the need to better explain this. Table 2 evaluates performance on a challenging subset of the VISOR validation data that we created, containing only objects or actions not seen during training. This directly tests the model's ability to generalize beyond learned statistical correlations. On this "novel" subset, CERES achieves a **60.0% mIoU⊕**, a significant improvement of **+4.7%** over the strong ActionVOS baseline, highlighting its robustness.
> 2. **Generalization to New Datasets (Table 3):** The evaluations on VSCOS and VOST are zero-shot generalization experiments. Our model is trained only on VISOR data and then evaluated directly on the VSCOS and VOST validation sets without any fine-tuning. These datasets feature different objects, actions, and visual distributions. Our consistent state-of-the-art performance here demonstrates a strong capability to generalize open-vocabulary scenarios.
>
> We appreciate the feedback and will revise Section 5 to more explicitly frame these results in the context of open-vocabulary and zero-shot generalization for the camera-ready version.

---

> > ### Comment · Area_Chair_ZdDn · 2025-08-05
> >
> > myER, please could you take a look at the author response above and whether it addresses any remaining concerns you have, e.g. integration ofdata statistics and depth information

---

> > ### Comment · Reviewer_myER · 2025-08-06
> >
> > Thanks for the authors' reply and extra analysis of my questions. Now I have a clearer understanding of CERES's ability in state-change scenarios and open-vocabulary settings. I have decided to raise my rating.

---

### Official Review · Reviewer_NduK · 2025-07-03

**Clarity:** 3
**Significance:** 2
**Originality:** 3
**Rating:** 4
**Confidence:** 4

**Summary:**

The paper proposes a causal inference-based framework, CERES, to address robustness issues in Egocentric Referring Video Object Segmentation (Ego-RVOS). Ego-RVOS aims to segment target objects involved in human actions based on language queries (e.g., "knife used to cut carrot") in first-person videos. CERES introduces dual-modal causal intervention, including a Linguistic Back-door Deconfounder (LBD) for language debiasing and a Visual Front-door Deconfounder (VFD) for visual confounding mitigation. Experiments on VISOR, VOST, and VSCOS datasets validate CERES's state-of-the-art performance. Ablation studies confirm the efficacy of each module, with theoretical proofs and implementation details provided in the appendix.

**Questions:**

See Weaknesses.

**Ethical Concerns:**

["NO or VERY MINOR ethics concerns only"]

**Final Justification:**

Overall, after weighing the authors’ rebuttal and the material they have promised to add in the revision, I find the paper well written, methodologically innovative, and experimentally sound. I therefore lean toward a borderline-accept recommendation.

**Limitations:**

See Weaknesses.

**Quality:**

2

**Strengths And Weaknesses:**

Strengths:
1. Standardized Presentation. Well-structured writing, clear figures, and intuitive framework diagrams.
2. Rigorous Technical Foundation. Complete theoretical derivations (e.g., MMSE optimality proof in the appendix) and transparent implementation details.
3. Practical Research Value. The focus on robust Ego-RVOS addresses a meaningful challenge in egocentric vision with real-world applicability.

Weaknesses:
1. Ambiguous Causal Rationale for Language Bias. Is the "Language Bias" solely defined as "certain object categories frequently co-occur with specific actions"? Is all knowledge of such co-occurrences harmful? Some correlations represent valid prior knowledge (e.g., "open" strongly associates with objects like "fridge" or "cabinet"), which can be beneficial for prediction. The paper lacks a clear distinction between harmful spurious correlations and potentially beneficial prior knowledge learned from data statistics. What specific types of "spurious correlations" does CERES target?
2. Insufficient Novelty in Core Methodology. The key components of the VFD (Depth encoder, Memory-bank) and LBD modules rely heavily on established techniques. The overall RVOS segmentation backbone shows minimal innovation. Moreover, the Attention-Linear Family (ALF) assumption for mediator fusion may constrain representation power. Exploring nonlinear fusion mechanisms (e.g., gating) is recommended, or at minimum, ablation studies comparing ALF to nonlinear alternatives should be included.
3. Incomplete Computational Analysis. Integrating the Depth encoder and Memory-bank (MAttn) likely increases computational cost. The paper lacks metrics quantifying this impact (e.g., inference speed/FPS, FLOPs, parameter count comparisons vs. baselines).
4. Inadequate Experimental Explanation. For Fig.4: It is unclear how depth information specifically resolves occlusion. Visualizations (e.g., attention maps or depth overlays) illustrating the role of depth in handling occlusion are needed. And depth maps from Depth Anything V2 can be error-prone. The paper does not analyze CERES's sensitivity to depth estimation errors. Experiments testing robustness under noisy/depth maps or with weaker depth estimators are crucial.
5. Inflexible & Dataset-Dependent Temporal Context. The fixed window size (W=5) for MAttn relies on a short-term stationarity assumption. Its performance (Fig. 5) appears highly dataset-dependent. This fixed window is likely suboptimal for sequences with long-term dependencies (e.g., objects leaving and re-entering the view). An adaptive window mechanism is strongly recommended instead of a fixed value.

---

> ### Author Rebuttal · Authors · 2025-07-31
>
> We sincerely thank you for your insightful feedback and constructive comments. We are especially encouraged that you acknowledged our work for its **rigorous technical foundation**, with **complete theoretical derivations** and **transparent implementation**; its **practical research value** in addressing a meaningful challenge in egocentric vision; and its well-structured and clear presentation.
>
> ---
>
> > `Weakness 1 / W1` Ambiguous Causal Rationale for Language Bias. Is the "Language Bias" solely defined as "certain object categories frequently co-occur with specific actions"? Is all knowledge of such co-occurrences harmful? Some correlations represent valid prior knowledge (e.g., "open" strongly associates with objects like "fridge" or "cabinet"), which can be beneficial for prediction. The paper lacks a clear distinction between harmful spurious correlations and potentially beneficial prior knowledge learned from data statistics. What specific types of "spurious correlations" does CERES target?
>
> Thank you for this excellent point.
>
> We agree that some statistical priors (e.g., ‘knives cut’) are beneficial. Our work targets the model’s **over-reliance** on these priors, a shortcut learning behavior that is especially harmful in Ego-RVOS. The core task is to determine if an object is actively involved in an action (aka. positive), and over-reliance on language priors causes failures on less common pairs (e.g., segmenting a 'knife' even if it's just lying on the counter, not being used for 'cutting').
>
> By de-biasing the text features, our LBD module dismantles these language-based shortcuts. **This prevents the model from defaulting to common pairings and instead necessitates a greater reliance on visual evidence.**
>
> To validate that LBD specifically addresses this issue, we performed a new ablation on the VISOR "hard" subset (rare concepts where shortcuts fail):
>
> Table: Ablation on the VISOR "hard" subset.
> |Method|IoU⊕|gIoU|
> |-|-|-|
> |ActionVOS|58.40|69.90|
> |+ LBD only|61.4 (+3.0)|71.8 (+1.9)|
> |+ VFD only|60.5 (+2.1)|71.3 (+1.4)|
> |CERES|62.3 (+3.9)|72.2 (+2.3)|
>
> This result demonstrates that LBD effectively addresses the targeted bias, which is also grounded in the statistical analysis of the dataset's co-occurrence bias detailed in **More Appendix D.1 of Supplementary Material zip**. We will revise the main text (Sec. 1 & 4.1) to state this critical distinction more explicitly.
>
> ---
> > `W2` Insufficient Novelty in Core Methodology. The key components of the VFD (Depth encoder, Memory-bank) and LBD modules rely heavily on established techniques. The overall RVOS segmentation backbone shows minimal innovation. Moreover, the Attention-Linear Family (ALF) assumption for mediator fusion may constrain representation power. Exploring nonlinear fusion mechanisms (e.g., gating) is recommended, or at minimum, ablation studies comparing ALF to nonlinear alternatives should be included.
>
> Thank you for this insightful comment regarding methodological novelty. We appreciate the opportunity to clarify our contribution and design choices.
>
> 1.  **Novelty is the Causal Framework, Not the Base Architecture:** We respectfully clarify that our core contribution is the novel **causal framework (CERES)** itself, designed to adapt powerful, pre-trained models to the unique biases of the egocentric domain. The use of established components is intentional, as our goal is to demonstrate how our causal interventions can robustly enhance existing SOTA models for this challenging data shift. Our consistent gains across multiple backbones (Table 1) support this claim.
>
> 2.  **Justification for Depth Fusion (DAttn):** We agree that the choice of fusion mechanism is critical. We opted for our Depth-guided Attention (DAttn) for two principled reasons:
>  - **Causal Grounding:** The Attention-Linear Family (ALF) assumption allows our DAttn to be a direct implementation of Pearl’s front-door adjustment (see Appendix A), making it a theoretically grounded choice, not an ad-hoc one.
>  - **Empirical Superiority:** In **Table 4 (row 3 vs. 4)**, we compare our DAttn against **a standard nonlinear MLP fusion (RGB+Depth)**. Our causally-guided DAttn (63.3% mIoU⊕) not only outperforms the MLP fusion (62.1% mIoU⊕) but also better discriminates against distractor objects (**15.8 vs. 17.5 mIoU⊖**).
>
> This result reveals a key architectural insight. A standard MLP performs "blind" fusion, lacking the structure to leverage depth. In contrast, our DAttn explicitly uses depth as a geometric query to attend to relevant visual features while actively suppressing distractors. This structured reasoning is something a simple MLP cannot easily replicate, explaining the superior robustness of our causally-motivated design.
>
> ---
> > `W3` Incomplete Computational Analysis. Integrating the Depth encoder and Memory-bank (MAttn) likely increases computational cost. The paper lacks metrics quantifying this impact.
>
> Thank you for this excellent point. We have added the requested computational analysis, which shows that our causal framework is a more efficient path to state-of-the-art performance than simply scaling the backbone.
>
> Table R1: Computational Cost vs. Performance on the VISOR dataset.
> (FPS measured on an NVIDIA 3090 GPU at 448x448.)
> |Method|Backbone|Param. (M)|FPS|mIoU⊕ (%)|
> |-|-|-|-|-|
> |ActionVOS|ResNet101|195|23.8|58.4|
> |ActionVOS|VSwin-B|237|15.4|62.9|
> |Ours|ResNet101 + DAv2-B|306|18.2|64.0|
>
> Crucially, our method outperforms the stronger ActionVOS (VSwin-B) baseline in both accuracy (**+1.1% mIoU⊕**) and speed (**18% faster**). This demonstrates that our method’s efficiency is rooted in its design, not just model size.
>
> While the separate depth encoder adds overhead, these results validate our approach and strongly motivate future work on integrating these causal principles into a single or more efficient encoder, potentially through knowledge distillation or structured pruning.
>
>
> ---
> > `W4` Inadequate Experimental Explanation. For Fig.4: It is unclear how depth information specifically resolves occlusion. Visualizations (e.g., attention maps or depth overlays) illustrating the role of depth in handling occlusion are needed. And depth maps from Depth Anything V2 can be error-prone. The paper does not analyze CERES's sensitivity to depth estimation errors. Experiments testing robustness under noisy/depth maps or with weaker depth estimators are crucial.
>
> Thank you for these insightful questions, which allow us to clarify the core mechanism and robustness of our method.
>
> 1. **On the Role of Depth in Handling Occlusion:**
>
> Our approach is founded on a key principle: occlusion severely degrades the semantic understanding derived from RGB features, but the object's spatial geometry remains far more stable when captured by a depth map.
>
> For example, when an object like a "knife" is occluded by a "hand," its visual semantics are confounded by the hand's appearance, leading to feature bias and segmentation errors. However, the depth map preserves the distinct spatial planes, clearly delineating the foreground occluder from the background object. Our Depth-guided Attention (DAttn) directly leverages this by using the stable depth features (Md) to guide attention over the less reliable RGB features (Mv). This implements a front-door adjustment, forcing the model to ground its segmentation in the object's consistent spatial location.
>
> To provide direct visual evidence, we will add the attention map visualizations you suggested to the appendix.
>
> 2. **On Robustness to Depth Estimation Errors:**
>
> This is an excellent point, and we agree it is crucial to validate. We have performed this exact analysis in **Supplementary Material zip file of More Appendix E.2 (Figure E.1)**.
>
> Our experiments show that CERES is highly robust to imperfections in the depth map. We tested this by corrupting the depth features with progressively stronger Gaussian noise. The results show that even with significant noise (std. dev. σn = 0.8), the performance degradation is minimal: **mIoU drops by less than 1% relative to the baseline (64.0% → 63.5%)**. This graceful degradation confirms that our DAttn mechanism effectively uses the general geometric structure and does not depend on pixel-perfect depth. We will add a sentence to the main paper to more clearly reference this appendix analysis.
>
> ---
> > `W5` Inflexible & Dataset-Dependent Temporal Context. The fixed window size (W=5) for MAttn relies on a short-term stationarity assumption. Its performance (Fig. 5) appears highly dataset-dependent. This fixed window is likely suboptimal for sequences with long-term dependencies (e.g., objects leaving and re-entering the view). An adaptive window mechanism is strongly recommended instead of a fixed value.
>
> Thank you for this insightful comment regarding our temporal modeling.
>
> Our use of a fixed window (W=5) was a deliberate design choice, representing a balance between model simplicity and strong performance on established Ego-RVOS benchmarks.
>
> We agree that an adaptive mechanism is more powerful for general, long-form videos. However, current standard datasets are not structured to support this; for instance, clips in the VISOR benchmark are typically sampled to an average of **10 frames with a 0.8-second interval**. This short-clip structure makes it very difficult to meaningfully train or evaluate a more complex long-term memory module. Our simpler fixed-window approach is thus well-suited and demonstrably effective for this data regime (as shown in Figure 5), avoiding unnecessary complexity.
>
> We concur that as long-form egocentric video datasets become available, developing an adaptive temporal model will be a crucial next step. Architectures like X-Mem [1] would be a promising direction for such future work, and we will clarify this motivation in our final manuscript.
>
> [1] Ho Kei Cheng, et al. X-Mem: Long-Term Video Object Segmentation with an Atkinson-Shiffrin Memory Model. In ECCV 2022.

---

> > ### Comment · Area_Chair_ZdDn · 2025-08-05
> >
> > NduK, please could you take a look at the author response above and whether it addresses any remaining concerns you have, e.g. ambiguous rationale for language bias

---

> > ### Comment · Reviewer_NduK · 2025-08-06
> >
> > Thank you for the detailed response. The rebuttal alleviates my concerns about the method’s novelty, and the analysis of computational efficiency is convincing. I am therefore inclined to raise my score to borderline accept.

---

### Decision · Program_Chairs · 2025-09-17

**Decision:**

Accept (poster)

**Comment:**

This paper received final ratings of 4 borderline accepts. Initially, the reviewers praised the paper for its practical nature and value; state of the art results across many backbones; and thorough evaluations. Regardless, the reviewers raised some issues in their reviews, namely: the definition of bias and how this was added into the causal framework and the motivation for including depth and whether this makes the results fairly comparable with previous methods. During the rebuttal stage, a lot of the time was spent discussing these two aspects, particularly on the need for depth and how biases were integrated. As a result of this, all reviewers ended with a positive recommendation towards the paper. The AC agrees with the reviewers' comments and final scores and sees no reason to overturn the majority decision for this paper and recommends acceptance for this paper.

The AC reminds the authors to add in the additional discussions from the rebuttal and update the camera ready with the promises that they made during the rebuttal.